# Lessons learned in coupling atmospheric models across scales for onshore and offshore wind energy

Sue Ellen Haupt[1], Branko Kosovic[1], Larry K. Berg[2], Colleen M. Kaul[2], Matthew Churchfield[3], Jeffrey Mirocha[4], Dries Allaerts[5], Thomas Brummet[1], Shannon Davis[6], Amy DeCastro[1], Susan Dettling[1], Caroline Draxl[3], David John Gagne[1], Patrick Hawbecker[1], Pankaj Jha[4], Timothy Juliano[1], William Lassman[4], Eliot Quon[3], Raj K. Rai[2], Michael Robinson[6], William Shaw[2], Regis Thedin[3]

[1] National Center for Atmospheric Research, Boulder, CO, 80301, USA.

[2] Pacific Northwest National Laboratory, Richland, WA, 99354, USA.

[3] National Renewable Energy Laboratory, Golden, CO, 80401, USA.

[4] Lawrence Livermore National Laboratory, Livermore, CA, 94550, USA.

[5] Delft University of Technology, The Netherlands [6] Wind Energy Technology Office, US Department of Energy, Washington, D.C., 20585, USA.

*Correspondence to:* Sue Ellen Haupt (haupt@ucar.edu)

**Abstract.** The Mesoscale to Microscale Coupling team, part of the U.S. Department of Energy Atmosphere to electrons (A2e) initiative, has studied various important challenges related to coupling mesoscale models to microscale models for the use case of wind energy development and operation. Several coupling methods and techniques for generating turbulence at the microscale that is subgrid to the mesoscale have been evaluated for a variety of cases. Case studies included flat terrain, complex terrain, and offshore environments. Methods were developed to bridge the *terra incognita,* that scale from about 100 m through the depth of the boundary layer. The team used wind-relevant metrics and archived code, case information, and assessment tools and are making those widely available. Lessons learned and discerned best practices are described in the context of the cases studied for the purpose of enabling further deployment of wind energy.

## 1. Introduction

Whether one is planning for where to deploy future wind farms, micrositing turbines within a wind farm, or designing optimal wind farm control, it is crucial to include the impacts of the large-scale (mesoscale, meaning thousands to hundreds of thousands of meters) flow as well as to model at the microscale (on the order of meters to tens of meters). As much of the energy of the atmosphere resides in the largest scales, correctly modeling those scales as well as the turbulence and energy dissipation at the microscale provides the most accurate picture of the flow and energy available for harvest.

The models for the two scales tend to be disparate, however. Although both sets of models are numerical discretizations of the Navier Stokes equations, they are built for different purposes. The mesoscale models are formulated for weather forecasting, have larger grid spacing over larger domains, and include parameterizations of many of the processes that are important for correctly modeling atmospheric flow, such as radiative transfer (short wave incoming and long wave outgoing), boundary layers, surface layers, cloud microphysics, land surface models, and more. Including such parameterizations is necessary to predict the flow accurately. Mesoscale models are also initialized with initial and boundary conditions from global models, which include the day-to-day weather fluctuations. On the other hand, microscale models are able to resolve details of terrain and wind turbines at a scale not available to the mesoscale models. But the microscale models do not include all of the atmospheric physics parameterizations of the mesoscale models. Thus, the solution to obtaining accurate flow prediction representing all relevant scales is to couple the mesoscale models to the microscale model.      .

Such coupling has long been a goal of modelers, but there have been a myriad of issues to work out. Some issues include:

- The mesoscale models are fully compressible while microscale models are typically incompressible or Boussinesq, where density differences are ignored except due to buoyancy.
- The gap between the typical resolutions of the two types of models – between about 100 m and traditionally 1000 m – known as the inner "grey zone" or the *terra incognita*, has been difficult to bridge (Wyngaard, 2004) - see section 2.1.
-  Treatment of surface conditions is often inherently different due to surface inhomogeneities that become important at the microscale  - see section 2.2.
- Best ways to couple the two models must be identified – see section 2.3.
- One must find ways to initiate turbulence at the microscale that is not resolved at the mesoscale - see section 2.4.
- Adding complexity, whether it comes from complex terrain or coupling atmosphere to ocean and wave models, complicates the picture and requires separate treatment - see section 2.6.
- Assessing how the models perform must be accomplished in the context of wind energy needs - see section 2.7.
- The uncertainty of the model results should be quantified to be most useful - see section 2.5.
- There is room for improvement in model parameterization – see sections 4.1 and 4.2.
- And finally, how can modern techniques such as improved parameterizations and machine learning be leveraged to improve modeling? See sections 4.2 and 4.3.

As part of the U.S. Department of Energy (DOE) Atmosphere to Electrons (A2e) initiative, the Mesoscale to Microscale Coupling (MMC) team was charged with studying these issues and more. The goal of the project has been to improve coupling between mesoscale and microscale simulations via enhanced guidance and new strategies

for setting up simulations and for the development of new tools that can be used across the community. This

philosophy recognizes that including the mesoscale forcing is critical to modeling the full energy transfer across

scales in the atmosphere. Specific objectives include:

•   Apply verification and validation techniques to the new modeling tools and develop estimates of the

uncertainty,

•   Reduce turbulence spin-up time in microscale simulations and hence decrease their computational cost,

•   Improve the surface layer treatment in microscale models to more accurately simulate wind speed and shear

over the rotor diameter,

•   Develop best-practice guidance for the community,

•   Prepare and document a suite of software tools that can be used across the community, and

•   Transition MMC research to the offshore environment.


Figure 1 illustrates the team's approach. The goal is to provide more realistic turbulence-resolving simulations
through coupling these scales. The team leveraged a case study approach to address these issues (Haupt et al.,
2019a). By working in the framework of studying particular situations for which we have observations, we can
better develop and assess tools to best match real-world situations, which is particularly important for studying
nonstationary meteorological conditions (such as frontal passages, thunderstorm outflows, baroclinic systems, and
low-level jets) or when considering changes of atmospheric stability associated with the diurnal cycle. In essence,
the objective is to have the microscale model "follow" the mesoscale model through dynamic changes while
appropriately modeling the fine-scale behavior of the flow. The approach is to select case studies from field
programs or observational data to identify challenging atmospheric conditions and test methods to simulate them.
Most of these datasets are from DOE-sponsored facilities in flat and complex terrain as well as from offshore sites.
The mesoscale modeling has focused on the widely used community model, the Weather Research and Forecasting
(WRF) model (Skamarock et al., 2008). Several microscale models have been tested, including the large-eddy
simulation (LES) version of WRF (WRF-LES) that can be run online where the inner nest derives the conditions
directly from the outer nest during the simulation, and several offline models, which are run after the mesoscale
model with inputs derived from those previous runs. Some aspects of the coupling that merit study include the
surface and boundary conditions, bridging the *terra incognita*, initializing turbulence at the microscale that is not
resolved at the mesoscale, the coupling methods themselves, and dealing with multiple sources of flow complexity,
including complex terrain, coastal flows, and offshore flows. The testing is grounded in rigorous verification and
validation configured specifically for wind energy plus uncertainty quantification, which emphasizes determining
parametric uncertainty of turbulence modeling in microscale simulations.

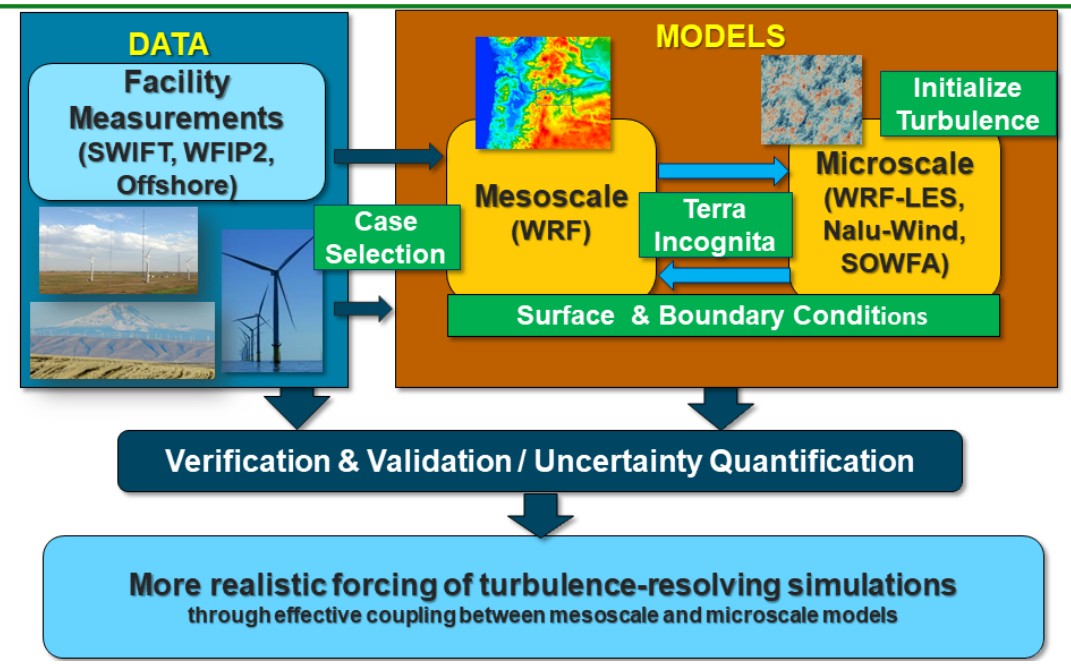

**Figure 1: The MMC team's case-based approach to addressing challenges of coupling the mesoscale to the microscale.**

An emphasis of the project is testing, evaluating, and comparing multiple methods of coupling the outer mesoscale flow to the microscale flow. Some methods use a single model (currently, WRF) at both scales, which ensures continuity across scales (internal coupling). Other methods incorporate forcing information from the mesoscale into a stand-alone microscale model (external coupling). This work is based on several preliminary investigations using WRF for both internal (Liu et al., 2011; Mirocha et al., 2014b; Muñoz-Esparza et al., 2014; Muñoz-Esparza et al., 2015) and external (Zajaczkowski et al., 2011; Gopalan et al., 2014) MMC, showing both promise and direction for future development. Rigorous comparisons of methods for different conditions and use cases provide insight into best practices. Another effort seeks to compare different methods of generating turbulence in the microscale models that is unresolved by the mesoscale forcing. The turbulence generation intercomparison was greatly facilitated by the development of Python-based assessment tools that are used via shared Jupyter notebooks. This effort includes design, testing, and deploying common code bases to simulate and assess the flows, which are now available on the public MMC GitHub (Quon et al., 2023a).

The team has archived simulation codes and model workflows for a range of case studies that can be used as a starting point for users to develop their own applications. Model codes, preprocessing, and postprocessing scripts are available on GitHub at (Quon, et al., 2023a,b,c, Gill et al., 2023, Hawbecker et al. 2023). Online documentation resides in a ReadtheDocs: (Mesoscale-to-Microscale Coupling, 2023). The goal of the code and workflow release is to promote high-fidelity coupled simulation capability to advance wind energy deployment through better

knowledge of the atmospheric conditions that drive energy harvest in wind farms. Modelers are invited to test our
models and workflows available at the GitHub site listed above.

This paper describes what we have learned about some of the difficult issues of coupling (Section 2), presents case
studies that were accomplished (Section 3), and discusses how enhanced methods, such as improved
parameterizations and machine learning, can help accomplish our goals (Section 4). Section 5 concludes with a
summary and a list of lessons learned plus suggests where future research should focus. Recommendations for best
practices are sprinkled throughout the paper.


**2    Some lessons learned**

The course of the research has investigated the topics laid out in Section 1, and here we summarize the work that has
led to lessons we have learned.

**2.1 The *terra incognita***

In coupled mesoscale–microscale simulations, including horizontal grid resolutions falling within the *terra incognita*
is almost inevitable. The *terra incognita*, coined by Wyngaard (2004), is the range of horizontal grid spacings where
turbulence models used in both mesoscale and LES do not work properly. The MMC project investigated the impact
of the *terra incognita* in coupled simulations (Rai et al., 2017; Rai et al., 2019). Our work suggests that the impact of
the *terra incognita* can be minimized using an appropriate choice of horizontal grid spacing, turbulence modeling
(dependent on the horizontal grid spacing), and grid refinement ratio (GRR) applied between the mesoscale to
microscale simulations. The most important consideration is that the horizontal grid spacing of the mesoscale
simulation should be at least comparable to the boundary-layer depth. Horizontal grid spacing smaller than the
boundary-layer depth produces erroneous structures in the simulated flow. Applying a GRR that allows simulations
to jump over the *terra incognita* not only alleviates the problem but also reduces the number of computational
domains. A larger value of GRR, however, also increases the fetch needed to generate turbulence on nested domains
due to the inertia of larger structures transported from the parent domain. The need for a larger fetch can be
mitigated by applying perturbations along the inflow boundaries of the domain (Section 2.4). In situations when the
GRR (between mesoscale and microscale domains) becomes large, it can be beneficial to use the LES three-
dimensional (3D) turbulence model (e.g., Smagorinsky, 1963) in the *terra incognita* region, provided that the
horizontal grid spacing is closer to 100 m, and then jump to grid spacing larger than the boundary-layer depth using
the GRR (Rai et al., 2019). However, the use of a 3D LES closure when the grid spacing is too coarse to resolve any
of the motions responsible for momentum transport can result in incorrect stress profiles, leading to significant
errors in wind speed within the ABL. The recently developed 3D planetary boundary layer (PBL) Mellor–Yamada
scheme (Juliano et al., 2022) fills a critical gap in this regard, providing for a consistent representation of transport at
scales finer than traditional mesoscale applications, but at scales too coarse to rely upon a 3D LES turbulence
closure (Section 4.1).
**2.2 Surface layer**
The surface layer (SL) traditionally represents approximately the lowest 10% of the atmospheric boundary layer
(ABL), within which the vertical fluxes of heat, momentum, and other constituents are assumed to approach nearly
constant distributions with height above the surface. Parameterization of the exchanges of these quantities between
the surface and the atmosphere within atmospheric models relies upon various SL scaling relationships, since the
vertical grid spacing in such models is generally too coarse to use a no-slip boundary condition. The particular SL
scaling employed, along with characteristics of the model spatial discretization, and the turbulence closure employed
to model turbulent exchanges above the surface, all interact to influence the application of the surface boundary
condition in atmospheric models, and subsequently impact resulting flow and other SL and ABL characteristics.
The most commonly employed SL scaling relationship used within atmospheric models is the Monin–Obukhov
similarity theory (MOST; Monin and Obukhov, 1954). MOST provides relationships to parameterize the fluxes
between the surface and atmosphere based on a small number of surface and near-surface atmospheric flow
parameters. While MOST is well established, relatively simple, and widely used, it is based on a number of
assumptions, including uniform terrain, horizontal homogeneity of both surface and atmospheric variables of
interest, steady flow and forcing conditions over time, and the appropriateness of ensemble-mean values of the
parameterized fluxes. These assumptions are reasonably well satisfied in most historical numerical weather
prediction and mesoscale atmospheric simulations, due in part to the use of coarse grid spacing, which satisfies the
appropriateness of ensemble mean representations within each grid cell, while also not resolving sharp transitions in
terrain features, horizontal heterogeneities, and meteorological forcing. However, the recent transition toward the
use of higher resolution in many mesoscale applications sharpens the representation of some or all of these features,
all of which increasingly violate the assumptions upon which MOST is based.
While the use of high horizontal resolution violates the applicability of MOST for one set of reasons, the use of high
vertical resolution can create additional problems, especially in settings for which a logarithmic mean profile shape
is not expected, such as within forest canopies or over significant surface waves or ocean swell. Moreover, care must
be taken not to place the lowest model grid cell too close to the surface.
Microscale atmospheric LES models also routinely apply MOST to formulate the surface stresses at each surface
grid cell based on the instantaneous time-varying horizontal velocities above. Even under highly idealized
conditions satisfying the assumptions of MOST in the aggregate, such models violate the appropriateness of the
ensemble-mean assumption.

Despite the above-mentioned caveats, MOST is still routinely applied in atmospheric simulations at all scales, owing
primarily to a dearth of alternatives. To improve its applicability, and the performance of simulating flow within the
SL more generally, numerous approaches have been developed, including various damping (Mason and Thomson,
1992) and correction factors (Khani and Porté-Agel, 2017); the use of more advanced turbulence subgrid-scale
(SGS) models (Bou-Zeid et al., 2005; Chow et al., 2004); taking care to properly set the computational mesh to have
the proper width-to-height ratio (Brasseur and Wei, 2010); and the use of additional near-wall stress
parameterizations (Brown et al., 2001) to distribute the surface stresses vertically. The impacts of many of these
methods on improving LES performance within the WRF model in wind-energy-relevant applications has been
examined in Mirocha et al. (2010), Kirkil et al. (2012), Mirocha et al. (2013), and Mirocha et al. (2014b).

SL modeling has also been extended to applications over forested landscapes for which a logarithmic vertical profile
of mean wind speed is not observed (see review by Patton and Finnigan (2012)). These methods are based on the
addition of momentum sink terms to the governing horizontal momentum equations to account for the increased
drag effects of foliage, with the magnitude of the drag expressed in terms of a leaf area index, which represents the
surface area of vegetation as a function of height. Modifications to elements of the SGS model, including eddy
viscosity coefficients and SGS turbulence kinetic energy (TKE), may also be included in such formulations.

Arthur et al. (2019) implemented the plant canopy model of Shaw and Patton (2003) into the WRF model and
demonstrated the ability of WRF-LES to recover expected distributions of winds and turbulence quantities in an
idealized plant canopy. Arthur et al. (2019) additionally combined concepts from the plant canopy approach and the
near-wall stress models used in various LES SGS formulations (Kirkil et al., 2012) to develop a novel distributed
drag implementation for the parameterized surface stresses. This model applies the expected surface momentum
stresses as drag terms in the horizontal momentum equations, distributed vertically over the lowest several model
grid cells. When applied in LES using the MOST surface boundary condition, this approach significantly improves
agreement between simulated mean wind speed profiles and their expected similarity relationships.

In addition to improving the implementation of MOST within atmospheric solvers, significant progress has also
been achieved in developing an alternative to MOST using machine learning (ML) to relate surface exchange to
relevant atmospheric and surface parameters obtained from observations. Details of this approach are provided in
Section 4.2.

**2.3 Coupling methods**

Over the course of this project, we have explored different frameworks for coupling mesoscale simulations to
microscale LES. Figure 2 depicts the various ways of classifying coupling strategies. Coupling approaches can be
classified according to the following properties: communication directionality (i.e., one-way or two-way coupling),
communication strategy (i.e., online through system memory or offline through file system), information transferred
(i.e., direct quantities such as wind speed, temperature, and surface fluxes, or indirect quantities such as tendencies
from the mesoscale budget), and the information transfer location (i.e., inflow/surface planes at the LES boundary,
or through the entire flow volume). A comparatively low-cost method for coupling mesoscale to microscale is via an
offline, periodic LES, which includes internal height-time varying source terms that provide mesoscale influence on
the microscale. For this approach, mesoscale simulation output is saved over a one-dimensional (1D) column at a
regular temporal interval (e.g., 10 minutes); this information is used with data assimilation techniques to force the
periodic simulation toward the desired mesoscale behavior. One way to achieve this forcing is through what we term
"profile assimilation," in which the microscale velocity and potential temperature solutions are plane-averaged at
each height at a given time. Those resultant mean profiles are compared with the desired mesoscale profiles, and the
difference is used to determine the amount of forcing required to drive the microscale mean vertical profiles to
match those of the mesoscale. One of the key lessons learned in this study is that with a strong forcing that enforces
the microscale mean vertical profiles to very closely match those of the mesoscale (what we term "direct profile
assimilation"), unrealistic turbulent fields sometimes form in the microscale simulation. This may be a natural LES
response to mesoscale profiles that are superadiabatic over too much of their vertical extent. To deal with this, we
developed a method that allows the microscale simulation more freedom to depart from the exact mesoscale vertical
structure (what we term "indirect profile assimilation"), but which will follow all the mesoscale trends in time
(Allaerts et al., 2020, 2023). Alternatively, the mesoscale forcing can be included by imposing time-height varying
source terms in the microscale LES. The forcing accounts for large-scale advection and the driving pressure gradient
and is extracted from the mesoscale simulation (Draxl et al., 2021). Any of these methods, though, assume a
horizontally homogeneous forcing field and are applicable only to homogeneous cases that are well-represented by
periodic boundary conditions. Although it is theoretically possible to apply an internal source term that varies three-
dimensionally in space to represent horizontally heterogeneous situations, we have not explored that approach;
however, others (Sanz-Rodrigo et al., 2021) have demonstrated the validity of that approach.  Instead, for
horizontally heterogeneous domains, or simulations that resolve turbines, we have focused our attention on
boundary-coupled simulations, which provide the highest degree of generality. Boundary-coupled simulations can
be conducted via online or offline coupling.

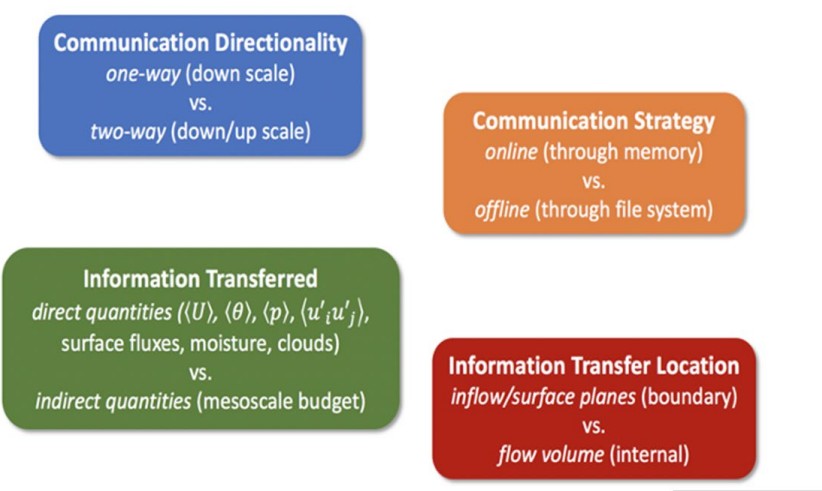


**Figure 2: Four ways of classifying coupling approaches.**

For offline coupling, the mesoscale output once again needs to be saved at regular temporal intervals to provide
boundary forcing for the LES. However, instead of 1D profiles, two-dimensional (2D) planes must be saved, which
increases the I/O and storage requirements considerably. Boundary coupling allows for simulation of a
heterogeneous domain for resolving complex terrain, mesoscale flows with significant horizontal gradients, or wind
farms.

Online coupled cases downscale from the mesoscale through nesting, usually within a single code; this allows for a
potentially streamlined workflow, as the downscaling usually involves setting runtime input parameters. Advantages
of an online coupled simulation is the ability to use consistent numerics and complete atmospheric physics across
spatial scales, as well as the ability to perform two-way coupling. However, because mesoscale meteorology models
are usually not developed with LES applications in mind, this coupling approach requires greater overhead and
poorly optimized parallelization of computing resources for the LES domain, imposing severe restrictions on the
ability to conduct large numbers of simulations. Note that a current DOE initiative focuses on development of
mesoscale (ERF) and microscale (AMR-Wind) models that are aimed at exascale HPC platforms. However, also
note that online coupling of mesoscale and microscale models that are based on the same formulation, i.e.,
equations, and use the same numerical discretization simplifies coupling and results in more consistent simulations
across scales.         Offline boundary-coupled simulations, however, are able to achieve higher simulation
throughput, which is crucial for parameter selection, sensitivity studies, or wind plant design applications. We
conducted a series of case studies directly comparing these approaches: one in a flat, fairly homogeneous onshore
environment (section 3.1, Allaerts et al., 2020; Draxl et al., 2021; Allaerts et al., 2023) and one in the offshore
environment (section 3.5, Thedin et al., 2022). Further case studies demonstrate the use of these techniques in
complex terrain (sections 3.3 and 3.4), resolving the coastal boundary (section 3.6), or in the offshore environment
with variable shallow water roughness and sea surface temperature (section 3.6).

We note that while the stand-alone microscale solver adds complexity to the setup, it allows for greater flexibility.
Most importantly, it allows for the study of the interaction of realistic weather conditions, complex terrain, and
turbines. The turbines can be coupled with aero-servo-elastic models using OpenFAST (2022 – see section 3.5.2)).
In the workflows presented in this paper, the turbine can be represented by actuator disk or actuator line models.
Note that the stand-alone, offline approach even allows the use of blade-resolved approaches.

**2.4 Initializing turbulence**

LESs are designed to explicitly resolve the energetically important scales of turbulence and the resulting fluxes and
transport those motions generate      within the flow. Models using grid spacings that are too coarse to resolve those
motions must instead rely on parameterizations (e.g., PBL schemes) to represent those processes. Therefore, when
forcing LES with mesoscale atmospheric data at the domain boundaries, either online or offline, a domain fetch is
required for the resolved scales of motion to appear within the LES flow field, since those motions are not resolved
within the inflow data. A similar issue is encountered when forcing LES with observations, as most observational
datasets do not contain sufficient spatiotemporal frequency to specify the turbulence field. In each of these cases, the
fetch required for resolved-scale turbulence motions to form and equilibrate to the large-scale forcing within the
LES domain can be extensive and represents a significant computational burden. The amount of fetch required
depends on multiple contributing factors, including surface roughness and terrain, wind speed, and atmospheric
stability. Generally, for a computation using specified inflow conditions during unstable conditions, the reduction of
fetch due to perturbations can be small, perhaps only around 100 grid cells in the direction of the mean flow.
However, during neutral or stable conditions, perturbation can foreshorten the fetch by several hundred grid points,
which can constitute a computational savings of 50% or more. Moreover, the flow field within the fetch will not
well represent either the mean or turbulence fields during the process of turbulence spin-up and equilibration.[1] To
ameliorate both the computational overhead and flow inaccuracies within LES forced in this manner, several inflow
perturbation methods have been developed and examined within the MMC project. These methods have been shown
to successfully promote the formation and equilibration of resolved-scale turbulence within LES driven by
mesoscale data and low-frequency observations, leading to substantial reductions of computational expense by
permitting the use of smaller LES domains while simultaneously improving the accuracy of the flow field beyond
the fetch. The inflow turbulence perturbation approaches that were examined within the project are briefly described
below.

---

[1] Within the fetch region, both the turbulence and mean flow statistics change rapidly, with turbulence developing, and the mean flow responding to those changes. Random perturbations applied just inside the inflow plane(s) produce uncorrelated gradients that, through the action of the governing equations, develop into robust turbulence features with expected correlations and energetics. During this process, there is often an associated reduction in mean wind speeds and a small change in wind direction near the surface, due to a temporary reduction in downward momentum transport -since the mesoscale closure is no longer providing that within the LES domain, and the turbulence within the LES domain has not yet developed the correlated structures responsible for downward momentum transport. The length of this region varies with stability and mean wind speed, with more stable and higher wind speeds generating longer transitional fetches. However, the mean and turbulence statistics of the flow do asymptotically approach their equilibrium values, after which no significant changes are observed with increasing distance from the inflow.

**2.4.1 Stochastic cell perturbation method**

The cell perturbation method (CPM) is based on the application of perturbed values of atmospheric temperature or velocity to "cells" (groups of contiguous model grid points in the horizontal and vertical directions) located just within the lateral edges of an LES domain (Muñoz-Esparza et al., 2014; Muñoz-Esparza et al., 2015; Mazzaro et al., 2019). Optimal choices for the amplitude, size and number of cells imparts variability upon the inflow that rapidly generates resolved-scale turbulence. Since the magnitude of the perturbation applied within each cell is drawn from a random distribution with a mean of zero, the method does not impose spatial correlations or turbulence structure explicitly. Rather, the mixture of random amplitudes and spatial correlations among the cells leads to the development of turbulence that is consistent with the large-scale forcing, defined by the ABL depth, surface roughness and temperature fluxes, and the distributions of mean winds and temperature – the latter contained within the inflow.

The CPM has been successfully applied in both idealized and real-data simulations for wind energy applications, including a diurnal cycle over an area of wind energy development in the U.S. Midwest region (Muñoz-Esparza and Kosovic, 2018), during a ramp event interacting with a parameterized wind farm in the Central Great Plains (Arthur et al., 2019), and in offshore resource characterizations in the North Sea (Thedin, et al. 2023) and U.S. East Coast regions (Hawbecker, et al., 2023), in each case showing improvement of the LES wind field, relative to unperturbed simulations

**2.4.2 Synthetic turbulence method**

Synthetic turbulence, such as the Mann method (Mann, 1998), are applied along the inflow boundaries of the LES domain to help generate realistic turbulence. The Mann synthetic method produces the turbulent winds in the three-dimensional volume, which is converted to a time series of inflow planes employing the frozen turbulence hypothesis. This method uses the spectral tensor of wave vectors to generate the isotropic turbulence and makes it anisotropic by applying the rapid distortion theory to the turbulent wind field. The inputs for controlling the variances of the turbulent field are the length scale and scaling intensity factor that controls the turbulent energy in the flow. If observations are available, we usually adjust the turbulence intensity by scaling the square root of the variances from the observations before applying it to the microscale model within the boundary-layer depth. Similarly, the frequencies of the turbulent inflow field at the domain boundaries can be adjusted based on the inflow wind speed. In addition to the Mann method, synthetic turbulence methods, such as TurbSim (Jonkman, 2006; Kelley, 2011; Rinker, 2018), can also generate turbulence along the inflow boundaries. Unlike the Mann method, TurbSim generates inflow planes in the time domain. If observations are available, the simulated turbulence can be forced to match an input time series and the structure of the turbulence can be controlled through empirical

coherence functions. These methods have been compared to CPM for flat terrain (Haupt, et al. 2019b, 2020) as well
as for offshore (see section 3.5).

**2.5 Quantifying uncertainty**

Modeling the atmosphere, at both meso- and microscales, is subject to uncertainty from a variety of sources.
Uncertainty propagates from the data used to specify initial and boundary conditions (e.g., reanalysis-based flow
fields, land surface properties, sea surface temperature data), from the form of model closures, and from specific
parameter values used within a closure. Sensitivities to these uncertain factors may display complex, nonlinear
interactions. Therefore, constraining the impacts on model predictions – particularly when considering mesoscale–
microscale coupled modeling – is difficult. A powerful, albeit computationally intensive, approach to evaluating
uncertainty in atmospheric model closures is to generate an ensemble of simulations that sample across a range of
parameter values. To adequately capture potential nonlinearities in the atmospheric model response, several dozen
or more ensemble members are typically required. However, once such a perturbed parameter ensemble is
generated, it may be extensively interrogated using a variety of meta-modeling techniques. For example,
Generalized Linear Models were used  by Yang et al. (2017, 2019) and Berg et al. (2019) for this purpose, while
Kaul et al. (2022) performed analyses using  Random Forest representations of the atmospheric model response .

In the context of wind energy applications, quantities of interest such as hub-height wind speeds, turbulence levels,
shear, and veer are known to generally show sensitivity to parameterizations of boundary layer turbulence and
surface fluxes, and these kinds of parameterizations have been most extensively targeted for uncertainty
quantification under the MMC project and related A2e projects. For example, uncertainty in mesoscale model
predictions over complex terrain owing to parameter values of PBL and surface schemes was examined by Yang et
al. (2017, 2019) and Berg et al. (2019). Reassuringly, these studies found that only a few parameters accounted for
most of the model uncertainty, although the identity of these parameters could vary diurnally and seasonally based
on the dominant state of atmospheric stability. Uncertainty owing to LES subgrid-scale turbulence closure
parameters in realistic mesoscale–microscale coupled simulations was examined by Kaul et al. (2022) and found to
trace predominantly to a single parameter (an eddy viscosity coefficient). However, the sensitivity of the modeled
flow to variations in this parameter was noted to vary significantly between two case studies with nominally similar
large-scale flow conditions but different smaller-scale flow structures (convective cells versus rolls), and to show
nonlinearity of response. For example, the hub-height wind speed showed much greater sensitivity to the eddy
viscosity coefficient, across the full range of  eddy viscosity coefficient values that were tested, in the case with roll-
type structures. TKE was also more sensitive in the case with rolls to changes in the coefficient value through the
lower half of the range of values tested. At higher values of the coefficient, turbulence was effectively damped, so
that the sensitivity of TKE to further increases in the coefficient became slight. In contrast, the case with a cellular
flow structure was better able to sustain turbulence, so sensitivity of TKE to the eddy viscosity coefficient persisted
across the full range of tested values, and sensitivities were greater at higher values of the coefficient.

Looking forward, much work remains to better characterize uncertainties within both mesoscale and microscale model predictions across a wider range of flow conditions, especially offshore. However, these initial studies give promising indications that uncertainty can typically be traced to a small number of model parameters and that the importance of these specific parameters can be interpreted in terms of flow physics considerations. Furthermore, application of meta-modeling techniques and leveraging machine learning approaches can greatly aid in detecting relationships and patterns within atmospheric model responses. Thus, efforts at uncertainty quantification not only meet a practical need to bound variability in atmospheric model predictions, but also can provide deeper insights to modelers that may ultimately drive improvements in parameterizations.

**2.6 Challenges of complexity and ways to approach**

Complexity comes into play in many manners for atmospheric flow. For the purposes of enhanced MMC for wind energy applications, we have focused on issues relating to complex terrain and offshore environments, including issues of correctly modeling atmospheric gravity waves but avoiding generating spurious ones.

**2.6.1 Complex terrain**

The coupling of mesoscale to microscale models using an offline approach (see Section 2.3) allows for the use of a stand-alone microscale LES solver, which brings the ability to use high-quality (in terms of mesh orthogonality) terrain conforming meshes. In complex terrain simulations, the assumption of horizontal homogeneity (often assumed in microscale simulations of the boundary layer) is no longer valid. Adding complex terrain to the simulation implies that periodic boundary conditions are not appropriate, and thus mesoscale coupling must be performed at the boundaries by means of spatiotemporal varying boundary conditions. A few additional complexities arise when performing this coupling.

To initialize the flow field in the microscale, the mesoscale solution is mapped onto the microscale domain. However, this mesoscale solution is obtained at a significantly coarser resolutions. In order to avoid unnecessary computational expense, a coarse grid must first be created to allow the mapping. After the mapping, further grid refinement should be performed to bring the domain to the desired microscale resolution. An additional terrain-conforming step must be taken to ensure the high-resolution LES grid is properly conformed to the underlying terrain elevation map. The boundary conditions that come from the mesoscale models only contain mean quantities, and thus the LES-resolved turbulence must be initiated in some way. Due to the inflow–outflow boundary conditions, two main strategies are used: the application of the cell perturbation method (see Section 2.4.1), or to allow the terrain itself to trigger the turbulence. We found that a perturbation technique is recommended because the terrain is only effective at generating the turbulence if it is sufficiently complex, in addition to significant fetch requirements (Hawbecker and Churchfield, 2021). For flat terrain Mirocha et al. (2014b) showed that under neutral

stratification fetch can be virtually infinite. An additional complication can be present in the mesoscale boundary
condition, where a single microscale boundary may experience inwards and outwards fluxes, and one must make an
appropriate choice of the boundary conditions for both the velocity and pressure, depending on the LES code of
choice. Finally, the terrain can trigger atmospheric gravity waves under certain stability conditions. The real
atmosphere extends for tens of kilometers vertically and infinitely horizontally, but a simulation domain is finite.
Atmospheric gravity waves reflect off of these domain boundaries and constructively or destructively interact,
creating spurious behavior. Approaches used to mitigate these spurious reflections and interactions are detailed in
Section 2.6.2.
**2.6.2 Atmospheric gravity waves**
As discussed in section 2.6.1, complex terrain can trigger atmospheric gravity waves, which microscale simulations
that include buoyancy effects will capture. In addition to complex terrain, atmospheric gravity waves can be
triggered by certain mesoscale weather patterns, land–sea interfaces, or wind farms themselves. The flow induced by
these atmospheric gravity waves can be of significant importance. But if these waves, whether significant or not to
the simulated problem, are allowed to reflect off of domain boundaries unchecked, they can cause spurious wave
interactions with unreasonable wave amplifications that completely pollute the rest of the flow. Our approach of
choice to mitigate spurious reflections is Rayleigh damping. Rayleigh damping is a simple but flexible concept. A
layer of some thickness is placed adjacent to a domain boundary in which a source term is introduced in the
momentum equation that forces the velocity toward a reference velocity with some time scale. Often we choose to
damp only the vertical velocity component to a zero reference state. However, Rayleigh damping is completely
general in that the reference velocity can be as complex as a 3D, time-varying field. Challenges with Rayleigh
damping include choosing an adequate thickness and proper time scale to effectively damp atmospheric gravity
waves. Too weak a damping layer will not completely damp reflected waves, but waves will reflect off too strong a
layer.  We suggest a damping layer thickness of 3-5 km with a damping time constant of 0.005 1/s, but additional
tuning likely will be required.  An additional challenge arises if the inflow boundary needs to be damped, which we
find to be the case in all inflow–outflow simulations, because upstream propagating atmospheric gravity waves must
be damped, but one does not want to damp incoming turbulence.
**2.6.3 The complexity of modeling offshore wind**
When switching from simulating complex terrain on shore to the offshore environment, our initial assumption was
that the problem became simpler. The offshore environment, due to a "flat" sea surface, seemed ideal for periodic
idealized simulations. Additionally, there are no heterogeneous surfaces to consider such as trees and cities, but only
water. This seemingly simpler problem turns out to be very complex and with fewer observational datasets to
compare against, meaning that it is very difficult to verify simulation accuracy. First, the ocean surface is generally
covered in waves of varying sizes, traveling in different directions, and with different periods. These waves have a
complex relationship with the atmosphere and ocean depth (see, for example, Jiménez and Dudhia (2018)) that
needs to be carefully considered in order to accurately simulate wind speeds within the boundary layer. Secondly,
sea surface temperature (SST) and SST gradients play an important role in determining the stability of the
atmosphere above. When considering SST gradients in simulations, we are often unable to utilize periodic boundary
conditions. Additionally, while many satellite-derived SST products exist and are used as the lower boundary
condition for temperature in a model, they are commonly only available once per day and rely heavily on gap-filling
techniques to produce estimates of SST where clouds have blocked their measurement, leading to biases in SST
datasets (Zuidema et al., 2016). These impacts may be more significant in the near-shore environment in which
offshore wind is focussed due to the occurrence of coastal upwelling, seasonal and climatological changes in ocean
currents such as the Gulf Stream, and the propensity for cloud coverage. Finally, there are also characteristics of the
offshore environment that are infrequently observed over land. Offshore low-level jets in the New York Bight –
where offshore wind plants are being developed – have been frequently observed to have jet noses below 100 m.
This means that the shear across the rotor will be extremely complex, as hub height for offshore turbines will be
above the jet nose. Another example is the propensity of extreme weather events in the offshore and coastal
environments. Hurricanes and other tropical disturbances commonly weaken as they move on shore due to increased
friction, or over colder seas, which reduces the latent energy that powers them. Such storms can remain quite strong
while located over warm ocean waters; however, the rate of storm motion can also play a role, as slower storm
movement can mix cooler water from below the thermocline up toward the surface, reducing the energy supply.
Upper level wind shear can also reduce the organization of the storm, leading to weakening or dissolution. All of
this leads to a very complex modeling framework requiring the coupling of ocean and atmospheric models (Shaw et
al., 2022).

**2.7 Wind energy relevant assessment and code availability**

To enable accurate assessment and repeatability of our science results, we have made all the essential components of
our studies publicly available. These components include (1) the problem definition, including data exploration,
curation, and transformation into useful simulation inputs; (2) the actual simulation inputs, including model
configuration files and scripts; and (3) postprocessing and synthesis of output. For this purpose, we have established
the A2e-MMC GitHub organization for archiving and disseminating our work archived at Quon, et all 2023a,b,c;
Gill et al., 2023; Hawbecker, et al. 2023. This public GitHub organization hosts Python analysis code, Python
analysis notebooks, code-specific input files, as well as our MMC-specific version of the WRF model that tracks the
community version (currently v.4.3), each constituting a separate version-controlled repository. For every study in
this project, the team has adopted workflows based on a common set of analysis and simulation codes within this
framework, thus ensuring apples-to-apples comparisons between results. To complement the technical content on
GitHub, we have also created a ReadTheDocs documentation site to provide an easily accessible high-level
overview of our project's accomplishments, describe our capabilities, and link to the resources on GitHub wherever
appropriate (Mesoscale-to-Microscale Coupling, 2023). We believe that in combination the GitHub and
ReadTheDocs will serve as a living record of the MMC project, as well as provide flexible and adaptable
documentation for future related projects.
**3    The value of case studies**
The team has developed and archived simulation codes and model workflows for a range of case studies that can be
used as a starting point for users to develop their own applications. The value of using a case study approach
includes the ability to choose real-world phenomena to model where observational data exist to validate our models.
That allows us to test different modeling approaches and techniques to discern which are most appropriate for the
particular situation. The cases that are curated are described briefly in the following sections, along with some
lessons learned for each.
**3.1 Flat terrain diurnal cycle**
To develop and test methods for coupling so that the microscale follows changes at the mesoscale, an early case
study of a diurnal cycle in flat conditions was chosen. This nonstationary case includes time-varying hub height
wind speed and direction, shear and veer, and turbulence intensity. For such a case, accurate downscaling of energy
from the mesoscale is important for predicting realistic turbulent flow features in the wind farm operating
environment.
Surrounded by grassland with no significant terrain changes within hundreds of miles, the Scaled Wind Farm
Technology (SWiFT) facility located in the southern Great Plains in West Texas forms an ideal flat terrain test site.
There are several meteorological measurement facilities near the SWiFT site hosted by Texas Tech University's
National Wind Institute (Hirth and Schroeder, 2014), including a tall meteorological tower and a radar wind profiler
with radio acoustic sounding system. In addition to the ideal terrain and availability of observational data, the site is
also chosen for its relevance to onshore wind energy installations in the United States. Details of the atmospheric
characterization are provided in Kelley and Ennis (2016).
From available data, the evening transition from 8 to 9 November 2013 was identified as a synoptically quiescent
diurnal cycle leading to nonstationary flow conditions at heights relevant to wind energy. The evolution of flow
parameters including wind speed, turbulence intensity, and virtual potential temperature follows a typical diurnal
pattern, featuring a morning transition, daytime convective boundary layer, afternoon/evening transition, and a
nocturnal low-level jet. The relatively simple geographical and meteorological conditions of the SWiFT diurnal
cycle make it an ideal case to study the performance of internal coupling methods throughout various atmospheric
stability regimes. The case has been used to evaluate existing coupling methodologies (Draxl et al., 2021) as well as
to develop new techniques (Allaerts et al., 2020, 2022). The WRF mesoscale simulation setup contains three nested
domains with 27 km, 9 km, and 3 km grid spacing, centered at the SWiFT site. The LES domains included 270, 90,
and 30 m resolutions.
Among the various lessons learned from this flat terrain diurnal cycle case, perhaps the most important one was
regarding the division of responsibilities between the mesoscale and the microscale solvers in an MMC framework.
The trends in the mean flow are set at the mesoscale level, and the microscale solver cannot correct for large biases
in mean-flow quantities or erroneous timing of large-scale events like the evening transition. The task of the
microscale solver is to fill in information on the unsteady, three-dimensional turbulent structures, which was often
accompanied by an improvement in the prediction of wind shear and mean turbulence statistics inside the boundary
layer, even in the relatively simple conditions of the SWiFT diurnal cycle. Further, the SWiFT case also highlighted
the need for more high-quality data extending up to higher altitudes for validation purposes. Despite the available
meteorological tower being taller than typically deployed towers, many boundary-layer processes with relevance to
wind energy take place above 200 m. For example, the low-level jet that developed during the SWiFT diurnal cycle
was predicted to attain its maximum wind speeds at a height between 250 and 350 m, but there was insufficient data
to validate this finding. Moreover, meteorological towers only present observations from a single column, which
means they cannot be used to assess how well the spatial variations in the turbulent flow fields are predicted. Note
that similar work has been carried out using data from the GABLS3 diurnal cycle case that included high-altitude
measurements to over 1000 m. Benchmark results are archived at Sanz Rodrigo et al. (2017a) with mesoscale to
microscale coupling results described by Sanz Rodrigo et al. (2017b) and archived in Sanz Rodrigo (2017b).
**3.2 Frontal passage causing a wind ramp**
A second case study (Arthur et al., 2020) leveraged MMC techniques to conduct simulations of a wind farm during a
frontal passage, for which rapid changes in wind speed, direction and temperature, and atmospheric turbulence were
observed. One of the key benefits of mesoscale–microscale coupling is the ability to examine wind energy
phenomena at the wind plant scale while resolving time-varying forcing from the mesoscale. The simulations
demonstrated the ability to capture the relevant mesoscale meteorological phenomena on a typical mesoscale
simulation domain, downscale those features to an LES domain containing a section of an operating wind plant,
represented as generalized actuator disks (GADs; Mirocha et al., 2014a), and simulate the interactions between the
time-varying meteorological flow and turbines, including wakes, power extracted, and turbulence phenomena. This
case study demonstrates the viability of fully online-coupled MMC simulations in WRF to address important issues
in wind plant behavior under realistic atmospheric operating conditions.
**3.3 Complex terrain case with high wind speeds and convective conditions**
The purpose of a first complex tterrain case study was to examine the flow structures near the surface, which depend
on many factors, including surface forcing. We investigated coherent structures present in the flow measured using
scanning lidar deployed near Wasco, Oregon, during the WFIP2 campaign (Wilczak et al., 2019) and those
simulated using WRF LES. The simulations utilized WRF to WRF-LES for the unstable condition case on 21
August and stable conditions on 14 August 2016 for the westerly flow. The model output was sampled in a way
consistent with scanning lidar data using plan position indicator scanning. We used the wind field of the innermost
domain that has a horizontal grid spacing of 10 m.

For both stability conditions, 90 east sectors, each 1 minute apart, were selected from the simulations and used to
compute the spatial proper orthogonal decomposition (POD) modes and energy (Berkooz et al., 1993). The actual
lidar data for the unstable case uses 49 east sectors with wind speed and heat flux values similar to those in the
simulations, 5–7 m/s and ~350 W/m$^2$, respectively. For the stable case, the actual lidar data employs 160 east sectors
with a wind speed of 10–12 m/s and heat flux ~$-30$W/m$^2$, similar to the simulated values. Figure 3 shows the spatial
POD modes 1 and 21 and the POD energy ( $\lambda$ , which denotes kinetic energy per unit mass of the flow) distributed
among many modes for the simulated and actual lidar data for two stability conditions. The first POD mode in all
cases shows the most significant coherent structures, followed by smaller structures for increasing mode numbers.
For the given stability conditions, the simulated and lidar cases showed similar shape and size variations for all
modes. The first few modes (modes < 5) show similar spatial structures in the POD modes for all stability
conditions. However, they exhibit different spatial structures for the higher POD modes. For instance, mode 21 in
the unstable case shows large open-cell-like structures, whereas mode 21 in the stable case shows streak-like
structures oriented in the predominant wind direction. This variation of flow structures in different modes can be
attributed to the forcing function. POD energy shown in Fig. 3 (right panels) depicts the turbulent energy associated
with each coherent structure starting from mode 2. The unstable conditions consistently exceed the POD energy (for
mode >1) in both simulated and observed lidar data. The cumulative energy (Fig. 3, inset) indicates that the first
mode of the stable condition case contains larger POD energy than the unstable condition case and requires larger
modes to represent the energy in the flow in observational data. Although the trend of varying POD energy shows
similarities between the two cases, the magnitude and the energy spread among the modes differ. Overall, the POD
modes of the different stability cases demonstrate that the simulations capture the important features of coherent
structures present in actual lidar data.


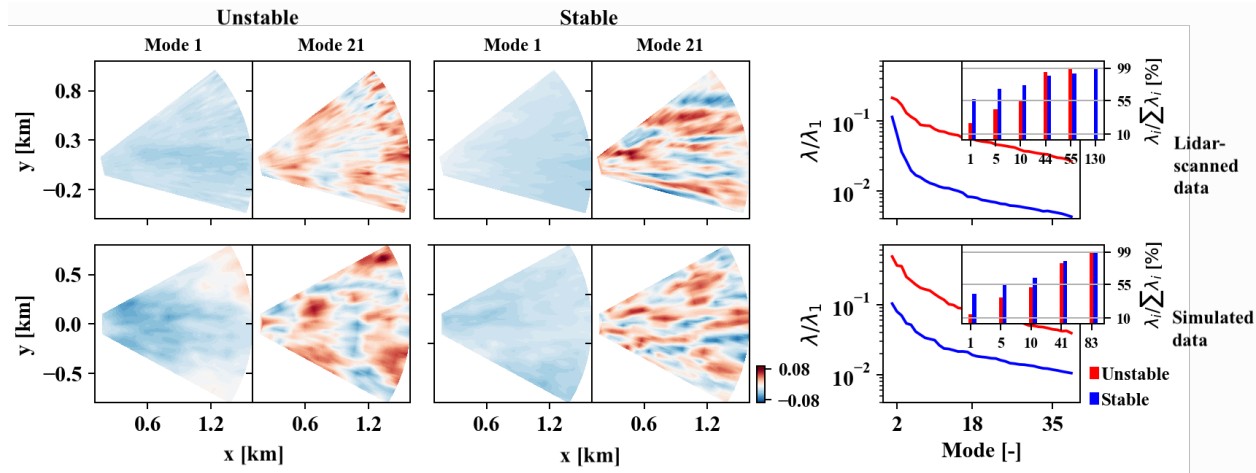

**Figure 3: Spatial POD modes 1 and 21 for the unstable (first and second columns) and stable (third and fourth columns) condition cases, and POD energy ($\lambda$) among the first several modes (fifth column) and their cumulative energy (in the inset). Panels in the top and bottom rows represent the results from observed and the simulated data, respectively.**

### 3.4 Complex terrain case using 3D PBL

This second complex terrain case also leverages measurements made during the WFIP 2 campaign, which covered many stability conditions, including cold air pools (CAPs) that tend to develop during synoptically quiescent periods. To study the ability of the 3D PBL scheme to capture such features, we chose a case from 10–20 January 2017 when a robust CAP was observed in the Columbia River Gorge. Such events are often challenging to represent accurately in mesoscale simulations due to the relatively small-scale boundary layer processes that must be parameterized. To better understand the spatial variability in meteorological and turbulence characteristics during the CAP lifecycle, we conducted WRF simulations following the High-Resolution Rapid Refresh (HRRR) reforecast configurations that were run for the WFIP2 project. For these simulations, the Mellor–Yamada–Nakanishi–Niino (MYNN; Nakanishi and Niino, 2006) scheme is run in the inner domain (horizontal grid cell spacing, $\Delta$ = 750 m) of a nested two-domain setup. A novelty of this study is the use of NCAR's 3D PBL parameterization (Kosovic et al., 2020; Juliano et al., 2022; Eghdami et al., 2022; Rybchuk et al., 2022), which was implemented into the WRF model for high-resolution mesoscale simulations. More information about the modeling setup and codes may be found at Mesoscale-to-Microscale Coupling, 2023.

Several key findings emerged from the WFIP2 CAP study, with additional details reported by Arthur et al. (2022). First, turbulence kinetic energy (TKE) measurements from the profiling lidar at the Gordon's Ridge site reveal that, compared to MYNN, the 3D PBL simulation more accurately represents the vertical and temporal variability in TKE. As a result, wind speed errors were lower in the 3D PBL simulation, especially during the CAP erosion period, which has been especially difficult to model (Adler et al., 2021). To better understand the leading cause of the improved performance by the 3D PBL compared with MYNN, we performed a sensitivity analysis using the 3D PBL scheme framework. More specifically, we modified the turbulence closure approach as well as the turbulent

length scale/closure constants formulation. The main reason for the improvement in TKE prediction is primarily
related to the different turbulent length scale/closure constants formulation. For 3D PBL simulations under
convective conditions, Juliano et al. (2022) reported similar findings regarding the primary importance of turbulent
length scale/closure constants formulation.

**3.5 Offshore wind case with a long offshore fetch**

The MMC techniques developed for onshore studies were tested for a first offshore scenario at the FINO1 research
tower, located in the North Sea. This case is representative of low roughness and low turbulence and leverages
measurements from the FINO towers and data from the Alpha Ventus wind energy plant.

**3.5.1 Comparison of coupling methods and turbulence generation methods**
Comparisons are made between members of an ensemble of mesoscale simulations, different coupling methods with
several models, and different turbulence generation schemes. The goal of the comparison is to assess the
performance of each approach and highlight their strengths and weaknesses. The approaches compared include:
● WRF to SOWFA using the indirect profile assimilation technique (IPA),
● WRF to SOWFA using the CPM at the inflow boundaries,
● WRF to WRF-LES without any added turbulence generation (control simulation),
● WRF to WRF-LES using the CPM at the inflow boundaries, and
● WRF to WRF-LES using the Mann model to generate the large-scale turbulence.

The domains used were 6 x 6 km, with the exception of SOWFA IPA, which had a 3 x 3 km extent. All cases have a
uniform 10-m grid resolution. Initial numerical experiments explored time-averaged vertical profiles at several
locations in the fetch to determine an appropriate size. Convergence of vertical profiles of turbulent metrics was
observed within a 3-km fetch distance. Thus, all the boundary-coupled scenarios considered were set up with a large
3-km extent fetch region to allow turbulence development. The results shown here represent the developed-flow
region, near the outlet boundaries. A qualitative visualization of the resulting flowfield is given in Fig. 4.

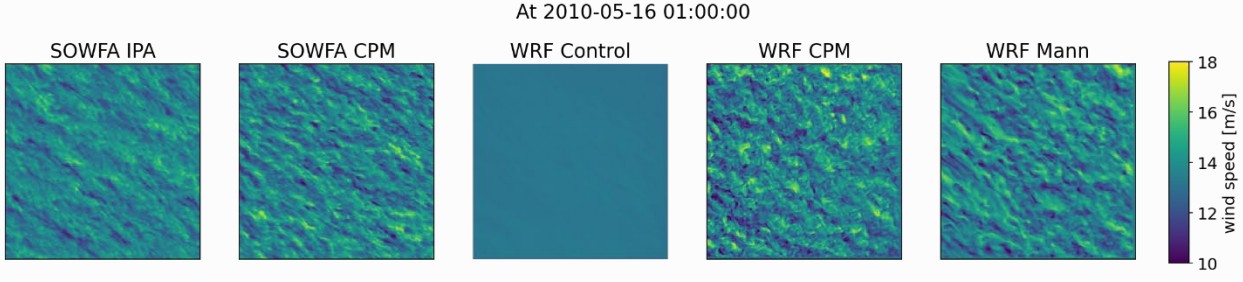


**Figure 4: Wind speed at 0100 local time on 16 May 2010 around the FINO1 location for the different methods**
**investigated. The original domains contain the fetch region. Shown here is developed-turbulence 3 x 3 km subdomain.**

Comparisons across the methods and observation data were made in terms of vertical profiles, power spectral
density content, correlations, and integral scales. Figure 5 shows the energy spectrum during one hour of the 4-hour
period of interest. The spectrum was obtained using 10-min Hamming windows with a 50% overlap. To obtain
smoother curves, we considered an ensemble average of several locations within the 3 x 3 subdomain shown in
Figure 4, leveraging horizontal homogeneity. WRF Mann and both CPM methods overestimated the energy content,
with the SOWFA IPA matching well the content with respect to observations up to frequency related to the LES
cutoff. The WRF control case showed very little content, as expected. The SOWFA IPA case is the only one where
the turbulence was not triggered by a numerical method, but rather developed using doubly periodic boundary
conditions. All of the vertical profiles are comparable, with the exception of the control simulation, which due to the
lack of resolved turbulence exhibited a larger shear profile. For a horizontal plane at 80 m, correlation maps were
calculated for every point      with respect to the central point, and correlation curves were obtained in the along-
wind and cross-wind directions. Taylor's hypothesis was observed to be valid for this case, by means of spatial
correlation and temporal autocorrelation. The correlation drop matched well the correlation from observations. The
correlations dropped to zero faster in the cell perturbation method cases for both SOWFA and WRF-LES, which
results in lower integral scales. Integration of the correlation curves yield the integral scales of the flow, shown in
Fig. 6.

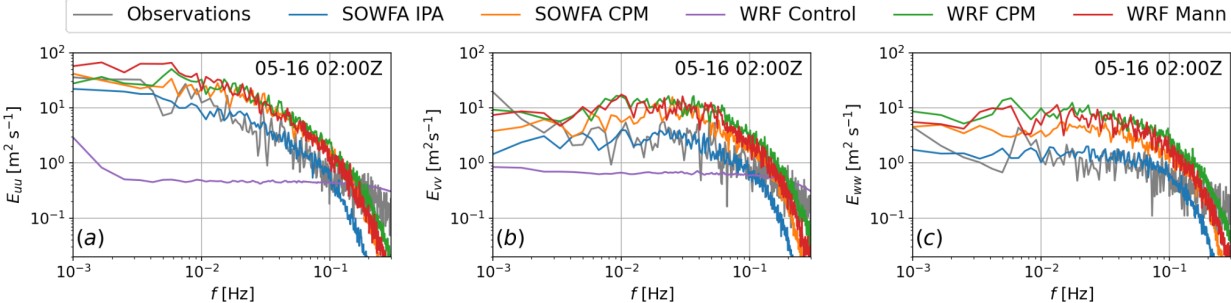


**Figure 5: Wind speed at 0100 local time on 16 May 2010 around the FINO1 location for the different methods**
**investigated. The original domains contains fetch region, showing only a developed-turbulence 3 x 3 km subdomain.**

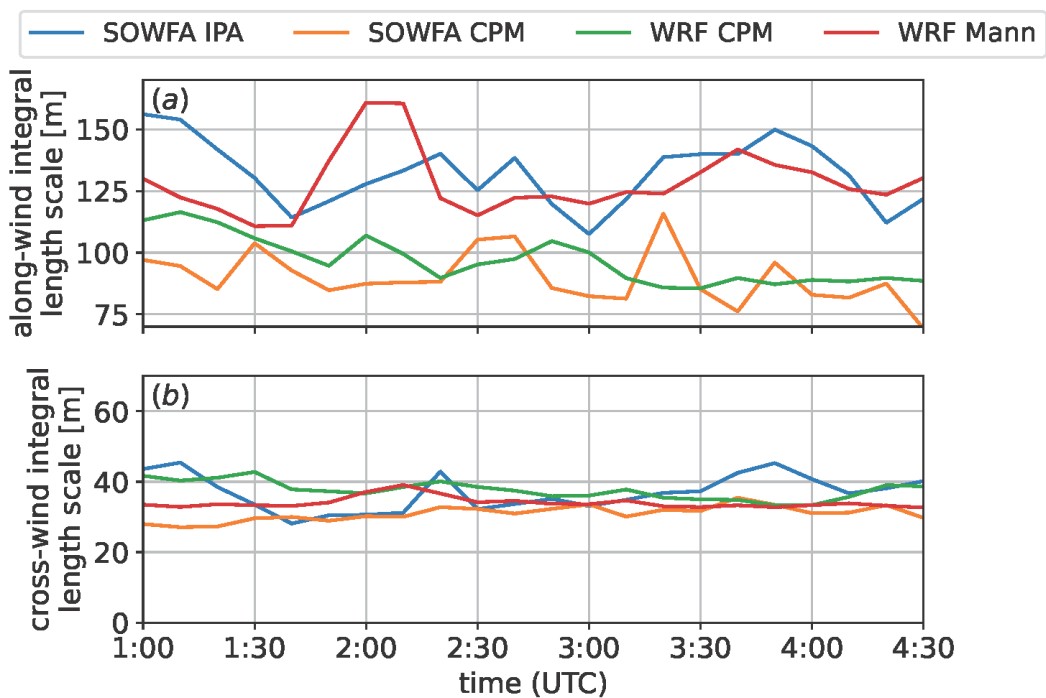

**Figure 6: Integral length scales calculated at 80 m in the along-wind and cross-wind directions for each coupling method.**

The integral scales present in the cases that used the cell perturbation method to generate turbulence are smaller throughout the interval of interest. That is likely a result of the way the perturbation method works, by imposing small-scale disturbances in the temperature field, thus triggering high-frequency, small-scale turbulence that does little to change the integral scales of the flow as a whole. The Mann method, on the other hand, imposes large-scale turbulence, and the LES resolves the smaller scales. The larger scales imposed on the field are clearly observed when comparing the integral scales of the flow to those obtained using perturbation methods. Lastly, the SOWFA IPA case resulted in integral scale values comparable to the Mann method in WRF-LES. For this SOWFA approach, the turbulence is developed by the use of periodic boundary conditions, which allows (in both space and time) the development of large-scale structures, ultimately resulting in long correlation fetches, and thus, large integral length scale values. While the SOWFA IPA domain was overall smaller, it was nonetheless able to resolve scales of the order of 150 m as shown in Fig. 6. The integral scales in the cross-wind direction were of comparable magnitude in all cases investigated.

**3.5.2 Alpha Ventus wind farm with generalized actuator disk – turbine comparison**

This section examines turbine wakes at the Alpha Ventus wind farm where the FINO1 tower is located and extends the analysis described in section 3.5.1. WRF to WRF-LES and WRF to SOWFA coupling approaches were extended to include a wind turbine parameterization using a GAD formulation (Mirocha et al., 2014a). We refer to them as WRF-LES-GAD and WRF-SOWFA-GAD, and each compares using CPM at the inflow boundaries vs. not adding

any turbulence. The time window of interest is a 2-hour window starting at 0100 local time (0000 UTC) on 16 May
2010. We consider a single turbine (AV10) for the purpose of this study.

Figure 7 presents a qualitative visualization of turbine wakes in the horizontal plane at hub height for the WRF-LES-
GAD approach. As in section 3.5.1, the LES domain is 6 km x 6 km with a horizontal grid resolution of 10 m, which
provides a large fetch as well as downstream distance for wake propagation. As expected, the simulation without
CPM does not resolve turbulence, and the resulting wake is what would be caused by an obstacle in the flow without
any mixing. The simulation with CPM includes resolved turbulence, and hence mixing in the shear region, leading
to a realistic wake. A comparison simulation using the WRF-SOWFA-GAD approach with CPM (not shown) also
concludes that modeling realistic wakes requires using a turbulence generating method.

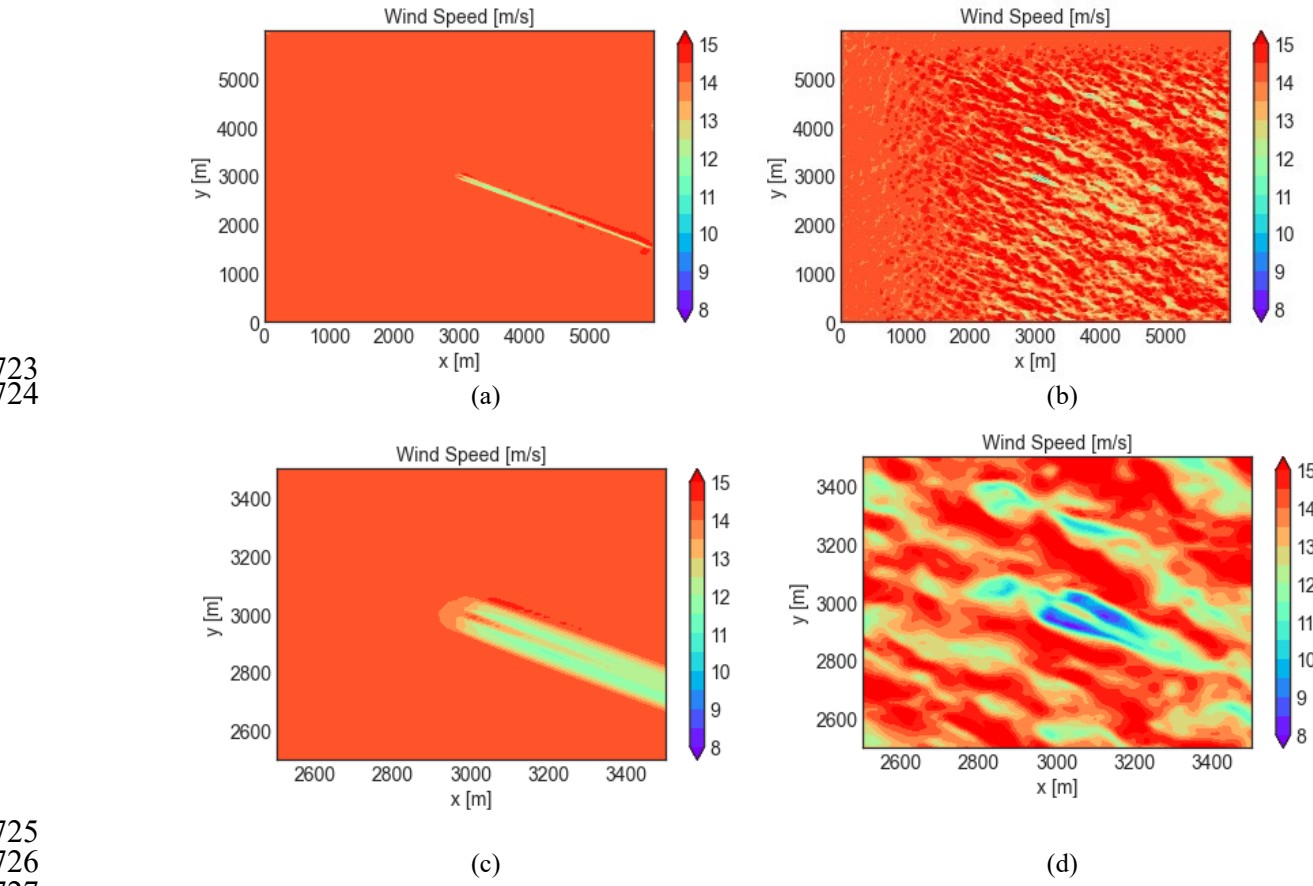

(a)                                        (b)

(c)                                        (d)
**Figure 7: Wind speed at 01:10:00 local time on 16 May 2010 in the domain containing the turbine (AV10) location using**
**the WRF-LES-GAD approach for (a) and (c) no CPM and (b) and (d) CPM. Entire domain is shown in (a) and (b). A**
**subset of the domain appears in (c) and (d).**


**3.6 Offshore Northeast U.S. coastal case**

A second offshore case is archived that studies the impact of different ways of representing surface roughness and
providing sea surface boundary conditions. The offshore environment in the Northeast United States is an active
area of research for wind energy development. Observations have recorded occurrences of persistent low-level jets
(LLJ) with jet noses commonly below hub height (Debnath et al., 2021). In this study we assess the sensitivity of
LLJ characteristics (e.g., jet nose height, maximum wind speed, low-level shear, etc.) to SST. We utilize six freely
available satellite-derived SST datasets from the Group for High Resolution SST website (Table 1 and Fig. 8) to
vary the lower-boundary condition of surface temperature in online WRF simulations.
**Table 1: Sources of SST datasets used in this study.**

| Dataset Source | Organization (year) | resolution (degrees) |
|---|:---:|---:|
| Naval Oceanographic Office (NAVO) | NASA, 2018 | 1 |
| Canadian Meteorological Center (CMC) | CMC, 2017 | 1 |
| Office of Satellite and Product Operations (OSPO) | OSPO, 2015 | 0.54 |
| Operation Sea Surface Temperature and Sea Ice Analysis (OSTIA) | UKMO, 2005 | 0.54 |
| GOES-16 | NOAA, 2019 | 0.02 |
| Multiscale Ultrahigh Resolution (MUR) | NASA, 2015 | 0.01 |

The simulations consist of five domains with grid spacing spanning from 6,250 m to 10 m. We used 88 vertical
levels with 20 m spacing below 1 km. We compare model results against observations from the New York State
Energy Research and Development Authority floating lidars. We assess model performance in capturing the LLJ
nose height, maximum wind speed, and low-level shear on each domain in order to compare how sensitive the
results are to SST on the mesoscale and microscale. With this comparison, we aim to determine whether model
sensitivity on the mesoscale translates directly to the microscale. In other words, can we expect the best performing
mesoscale model setup to be the best setup on the microscale?
Results indicate that ensemble mean error and spread for various characteristics of the offshore LLJ vary between
the mesoscale solutions and microscale solutions. However, variance within the microscale domains (domains 4 and
5) is small. Ensemble mean error, $\text{EME} = \sqrt{(s_o - \overline{s})^2}$ where $s_o$ is the observed quantity and $\overline{s}$ is the ensemble
mean) and bias of the low-level shear, hub-height wind speed (assumed to be at 118 m in this case), and jet nose
height vary across scales from mesoscale to microscale (Fig. 9). Additionally, the best mesoscale performer did not
lead to the best microscale performing setup in this case when considering these metrics. On the mesoscale, the
shear produced in the lowest levels was lower than what was observed. The LES results improved upon the low-
level shear but overcorrected the lowest level wind speeds and produced values lower than what were observed. It is
suspected that using a drag force locally consistent with MOST within the heterogeneous microscale simulation is
the root cause of this overcorrection of low-level winds. Future work must focus on generalizing this finding in
order to determine if mesoscale simulations can inform performance on the microscale prior to running simulations.

**Figure 8: Sea surface temperature datasets of varying resolution used as initial and surface boundary conditions over**
**water.**

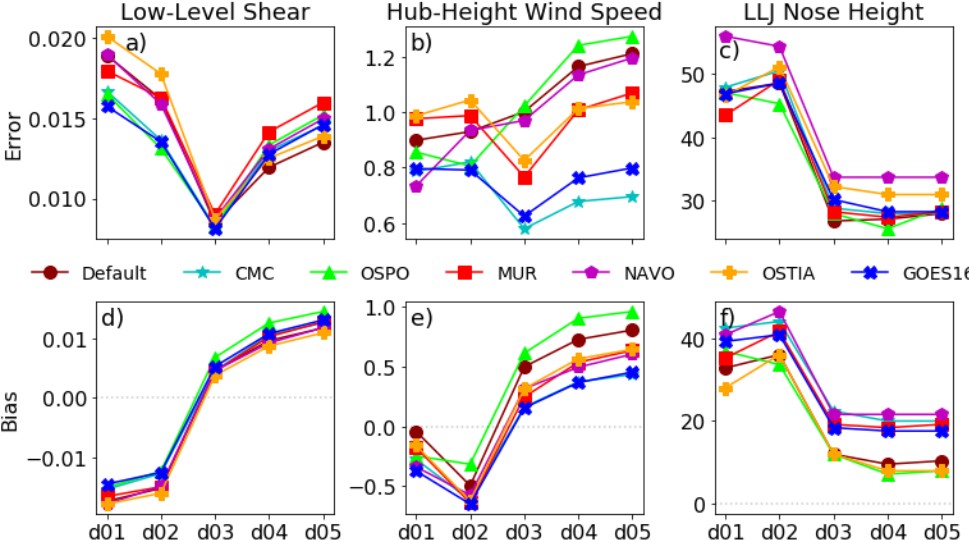


**Figure 9: Error (top) and bias (bottom) for each case on each domain for low-level shear (left), hub-height wind speed**
**(middle), and LLJ height (right). Units for error are a) and d) s$^{-1}$, b) and e) m s$^{-1}$, c) and f) m.**


**4    Contributions of enhanced methods**

The MMC team additionally tested ways to improve the models both in terms of improved physics as well as to test
the efficacy of machine learning methods.

**4.1 Three-dimensional planetary boundary layer parameterization**

Traditional PBL schemes in mesoscale models are one-dimensional – that is, they parameterize only the vertical
turbulent mixing under the assumption of horizontal homogeneity. In this sense, the vertical turbulent fluxes of
momentum ($<u'w'>$ and $<v'w'>$), potential temperature ($<\theta'w'>$), water vapor mixing ratio ($<q_v'w'>$), and any
other relevant scalars ($<\varphi'w'>$, where $\varphi$ is a scalar variable, such as cloud water mixing ratio) are computed. By
definition, the horizontal homogeneity assumption neglects horizontal gradients in resolved quantities, as well as
the vertical gradient in vertical velocity. Therefore, the vertical turbulent fluxes are dependent on only vertical
gradients. However, this assumption is not justified at model resolutions in the *terra incognita* ($\Delta \approx 100$–1000 m),
where turbulence is partially resolved, and thus, horizontal gradients play an important role (e.g., Kosovic et al.,
2021). A main consequence of ignoring horizontal gradients in the *terra incognita* and under convective conditions
is the development of spurious structures (termed modeled-convectively-induced secondary circulations, or M-
CISCs, by Ching et al. (2004)], which can have a deleterious effect on the model solution. Furthermore, most 1D
PBL parameterizations rely on the 2D horizontal diffusion scheme of Smagorinsky; however, this scheme was
originally introduced for numerical stability and is therefore not physically motivated (Smagorinsky, 1990).

To address the fundamental research challenge of modeling in the *terra incognita*, our team has implemented the
3D PBL parameterization of Mellor and Yamada (Mellor, 1973; Mellor and Yamada, 1974; Mellor and Yamada,
1982) into the WRF model. This new parameterization does not impose the assumption of horizontal homogeneity;
thus, it considers both vertical and horizontal gradients when computing all six momentum stresses and the full
tensor for scalars (namely, $\theta$ and $q_v$), in addition to all components of the flux divergences. As a result, this
approach does not require the use of Smagorinsky's 2D horizontal diffusion scheme and shows promise at grid
resolutions in the *terra incognita*, especially under convective conditions. To examine the influence of accounting
for horizontal gradients, we set up different idealized model configurations under convective conditions and at
high-resolution mesoscale grid spacing ($\Delta = 250$ m). This grid spacing is considered to be mesoscale resolution
because it is not fine enough to fully resolve the most energetic eddies (i.e., the LES limit) due to the model's
effective resolution. The three single-domain, doubly-periodic configurations are: homogeneous surface forcing
(rolls and cells), sea breeze front initiation, and mountain–valley circulation. Results clearly depict the suppression
of M-CISCs by the 3D PBL scheme compared to a traditional 1D PBL scheme (Juliano et al., 2022). The impact of
the turbulent length scale/closure constant's formulation is found to be very important, such that M-CISCs may be
present in the 3D PBL solution when the length scale is insufficiently large and thus vertical mixing is not strong
enough. In general, we believe that the 3D PBL parameterization has potential to be useful both as a mesoscale-
only approach and as part of a mesoscale-microscale coupling strategy.

**4.2 Machine learning surface layer scheme**

Specifying lower boundary conditions in numerical simulations of high-Reynolds-number atmospheric boundary
layer flows requires estimating turbulent fluxes of momentum, heat, moisture, and other constituents. However,
these fluxes are not known *a priori* and therefore must be parametrized. Parameterization of surface fluxes in
atmospheric flow models at any scale, from global to turbulence-resolving large-eddy simulations, are based on
MOST where atmospheric stability effects are accounted for through universal, semi-empirical stability functions.
The stability functions are a function of the nondimensional stability parameter, a ratio of distance from the surface
and the Obukhov length scale z/L (Monin and Obukhov, 1946). However, their functional form is determined based
on observations using simple regression that cannot represent the surface-layer structure and governing parameters
under a wide range of conditions. We have therefore developed and tested a neural network (NN) ML model for
surface-layer parameterization (McCandless et al., 2022). We trained and tested the ML model using long-term
observations from the National Oceanic and Atmospheric Administration's Field Research Division tower in Idaho
and the Cabauw mast in the Netherlands. The offline comparison of MOST and the NN model surface-layer
parameterizations with observations from the Cabauw mast are shown in Fig. 10. We then implemented the ML
model in the FastEddy GPU-native LES model (Muñoz-Esparza et al., 2022) and the WRF single-column model.
The ML model implementation in Fast-Eddy demonstrates that it can accurately capture the diurnal evolution of an
atmospheric boundary layer as shown in Fig. 11.

The ML model implementation in the WRF model was tested using a single-column model (SCM) based on the
GABLS III intercomparison study case defined by Bosveld et al. (2014). The comparison of SCM simulations using
the ML model surface-layer parameterization with observations and the MOST parameterization demonstrates that it
can capture well the sensible heat flux, the skin temperature, the surface friction velocity, and the planetary
boundary layer height, but underestimates the latent heat flux (Fig. 12).

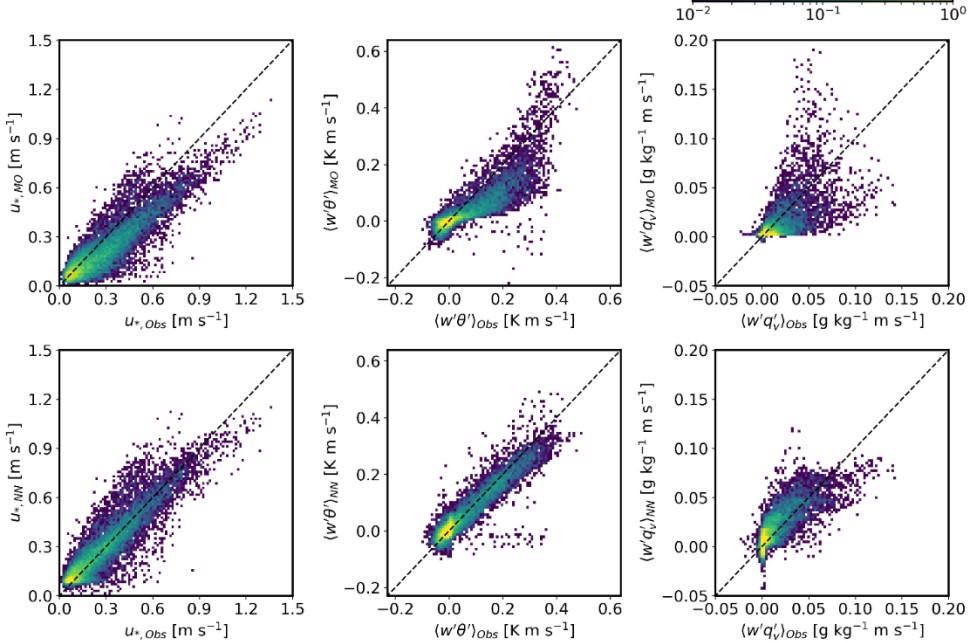


**Figure 10: Comparison of the MOST (top row) and an offline NN model (bottom row) surface-layer parameterizations of**
**surface friction velocity (left panels), sensible heat flux (middle panels) and moisture flux (right panels) with observations**
**from the Cabauw mast. Figure originally appeared in (Muñoz-Esparza et al., 2022).**

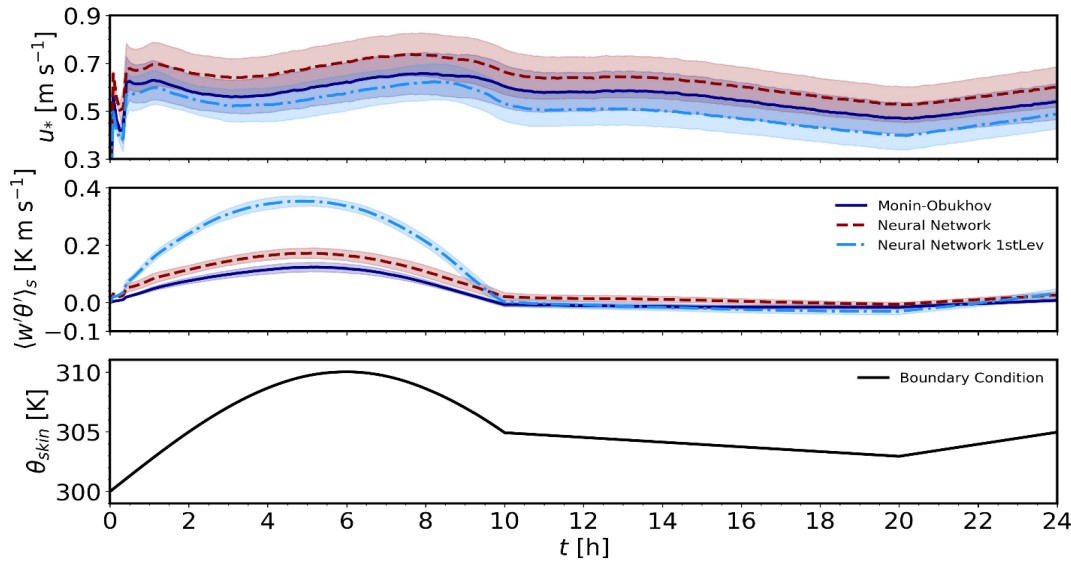

**Figure 11: Comparison of the diurnal evolution of an ABL using the FastEddy LES model with the MOST and NN model**
**surface-layer parameterizations: surface friction velocity (top panel), sensible heat flux (second panel), moisture flux**
**(third panel), and boundary forcing from surface skin temperature (bottom panel). The shaded areas show 1 standard**
**deviation from the mean over the simulation domain. Figure originally appeared in (Muñoz-Esparza et al., 2022).**

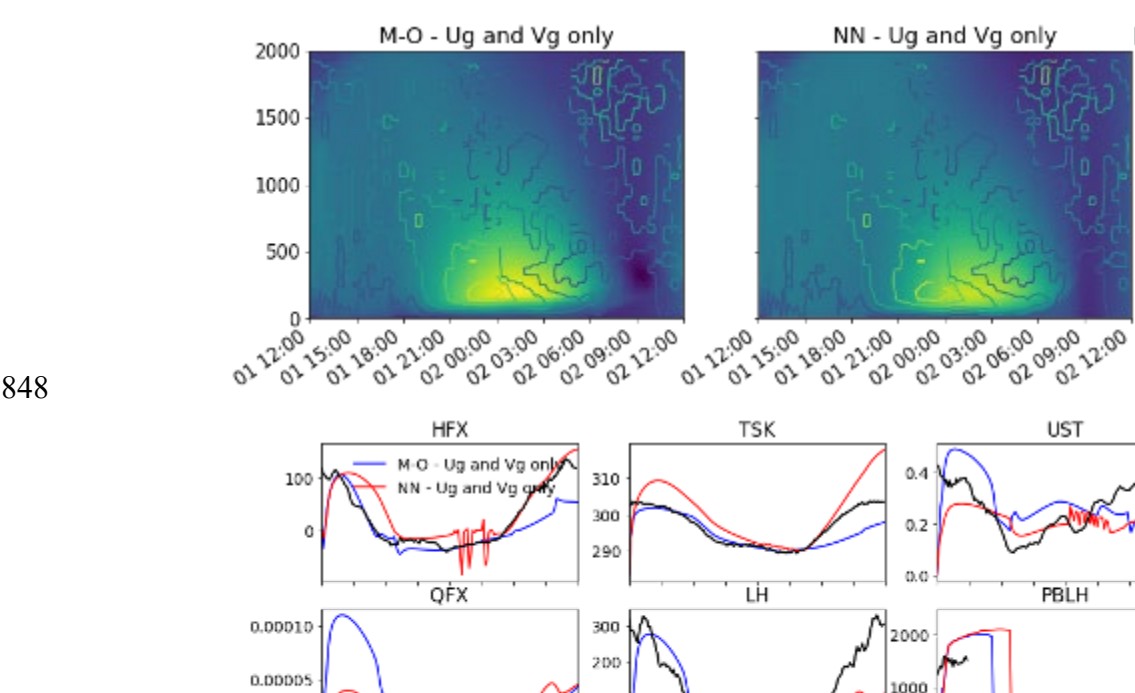



**Figure 12: Output from the SCM simulation of a GABLS III intercomparison study case using an idealized WRF**
**model. The figure compares WRF simulations using MOST and a neural network parameterization. The black line shows**
**the observed data from GABLS III (Cabauw) for comparison. "Ug and Vg only" refers to the single column simulations**
**only being forced by changes to the geostrophic wind. The bottom portion of the figure shows heat flux (HFX), skin**
**temperature (TSK), u\* (UST), moisture flux (QFX), latent heat (LH) and PBL height (PBLH).**

A potential reason for discrepancies between the ML model-predicted and observed latent heat flux is that the ML
model for the surface-layer parameterization implemented in WRF interacts with a land–surface model, which is
based on MOST.

The ML model for surface-layer parameterization demonstrates the potential to provide better estimates of surface
fluxes in comparison to commonly used MOST-based parameterizations. However, to develop a generally
applicable ML model it must be trained using long-term, consistent, complete, and quality-controlled observations
from a wide range of environments. Future research could focus on expanding the training dataset and testing the
model in mesoscale simulations over diverse locations.

**4.3 Downscaling with deep learning**

Microscale simulations, like the WRF-LES (30 m) generated over the Columbia River Basin for the Wind Forecast
Improvement Project 2 (WFIP 2), are able to model the very complicated flow associated with complex terrain
including downslope flows, mountain wakes, mountain–valley circulations, gravity waves, cold pools, and gap
flows. However, such simulations are currently too complex to configure and computationally expensive for use
outside the scientific research community. Here we tested using deep artificial neural networks on the LES to
directly downscale from mesoscale to microscale in complex terrain. Once trained, deep learning models can
generate high-resolution simulations from a coarse image in just a few seconds from mesoscale input. In addition,
we wished to demonstrate that the deep network models can then potentially be applied to regions other than the
LES domain on which they were trained.

We created high-resolution/low-resolution training sample pairs by subtiling relevant vertical levels of the LES on
the eastern portion of the domain and coarsening the tiles with average filters. We trained two separate Enhanced
Super Resolution Generative Adversarial Networks (ESRGANs; Ledig et al., 2017; Wang et al., 2018) to
accomplish the downscaling by training one GAN to downscale from 960 m to 240 m and the second GAN to
downscale from 240 m to 30 m, and applying the models successively. We set aside data from every third time step
in the LES for testing. Visually, the performance of the compound GAN architecture on testing data samples and the
larger domain was impressive (Fig. 13). We performed statistical analysis of the high-resolution GAN-generated
wind and compared it with the LES, finding good agreement in the power spectra, velocity gradient distributions,
and wind speed and wind direction distributions (Dettling et al., 2022). We found high Pearson correlation
coefficients and very low mean bias between the tiles of GAN-generated wind components and LES, as well as good
agreement in the moments of GAN-generated wind components with the LES, even in the higher-order moments,
skewness, and kurtosis (Dettling et al., 2023).

To demonstrate the potential of transfer learning, we extended the testing sample set to include the western half of
the WRF-LES, which contains part of Cascade Range including Mt. Hood. The western region is not only very
unique when compared to the training region in the east, it is also topographically much more complex. We
performed the same statistical analysis to compare the GAN-generated wind to the LES in the transfer learning
region and the results were encouraging (Dettling et al., 2023).

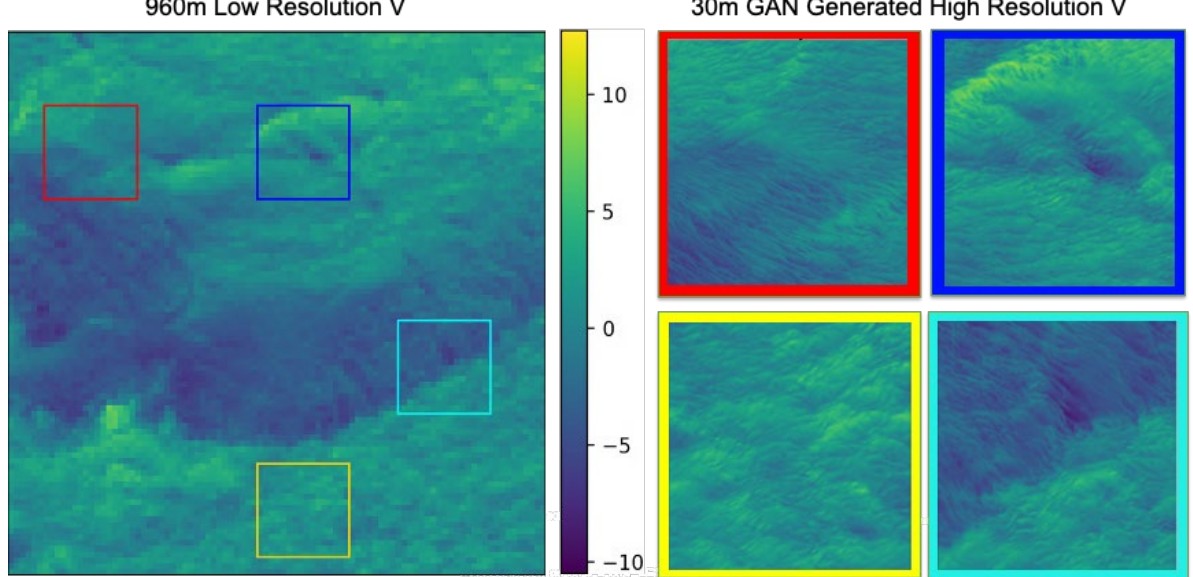

**Figure 13: Example of using the GAN to downscale from a coarsened 960 m resolution simulation (left image) to four example panels showing high-resolution 30 m generated images. The colors overlaid on the left panel correspond to the same color outlined image on the right panel.**

## 5    Conclusions

We have summarized the results of the U.S. Department of Energy (DOE)-sponsored Mesoscale to Microscale Coupling (MMC) project that has focused on the best ways to couple the mesoscale to the microscale in order to better understand and model the transfer of energy from the largest scales of the atmosphere to those scales that directly affect harvesting that energy via wind turbines. The approach of using case studies based on observations has been a productive approach to test methodologies and has kept the findings grounded in real-world atmospheric behavior. The approach has required that we choose progressively more difficult cases, bringing in real-world complexity to better understand the implications of that complexity and how to best model it. We have studied how the mesoscale setup impacts the microscale results, applying consistent and appropriate boundary conditions, multiple methods of applying the coupling between scales, bridging the *terra incognita*, initializing turbulence at the microscale that is not resolved at the mesoscale, and applying these methods in complex terrain and in coastal and offshore environments. We additionally explored improving model parameterization (3D PBL and a ML-based surface layer model) plus demonstrated deep learning methods for downscaling from mesoscale to microscale. It is important to apply assessment metrics that are most appropriate for uses in wind energy, considering more than merely mean winds, but also sheer, veer, turbulence intensity, and turbulent kinetic energy via metrics such as energy spectra, pdfs along the flow, covariance, and proper orthogonal decomposition.

Some specific lessons learned include:

• Microscale simulations cannot necessarily improve matches to measurements if forced with an inaccurate
mesoscale simulation (section 3.1).
• Idealized simulations may not well represent real-world phenomena and may be more difficult to initialize
well than real cases.
• Microscale data assimilation (through profile assimilation on a periodic domain) requires an approach that
allows the microscale to deviate from the mesoscale, otherwise wind and temperature profiles may not be
in the correct equilibrium, resulting in unrealistic turbulence (Allaerts et al., 2020, 2023).
• High-quality potential temperature profiles, in addition to wind profiles, are necessary when performing
microscale data assimilation with observational data (Allaerts et al., 2023; Jayaraman et al., 2022; Quon et
al., 2022).
• Accurately capturing transitional atmospheric boundary layers and intermittent stable boundary layers
remains a challenge (Allaerts et al., 2022; Quon et al., 2022).
• Without coupling across scales, even mesoscale flow is underresolved (Rai et al., 2019).
• Proper orthogonal decomposition analysis clearly indicates that the microscale contains energetic modes
that originated from the mesoscale flow (Rai et al., 2019).
• The upper limit of the *terra incognita* is the boundary layer depth, indicating that horizontal spacing
smaller than that (but larger than about 100 m) is likely to result in spurious secondary structures (Rai et al.,
2019).
• Spurious roll features from the *terra incognita* can translate into unrealistic flow in the microscale (Rai, et
al., 2019).
• Turbulence generation methods are necessary to avoid long fetches in developing turbulence at the
microscale that is not resolved at the mesoscale (Section 2.4).
• Temperature perturbation methods create turbulent fields with artificially small integral scales (Section 3.5)
• Uncertainty can typically be traced to a small number of model parameters and the importance of these
specific parameters can be interpreted in terms of flow physics considerations (Section 2.5).
• Certain conditions, such as complex terrain, can force gravity waves that reflect off of boundaries and grow
to spurious amplitudes. Such gravity waves can be mitigated by Rayleigh damping (Section 2.6.2).
• The best mesoscale simulations don't always translate to the best match to wind-relevant metrics for the
microscale simulation (Section 3.6).
• A three-dimensional planetary boundary layer can alleviate M-CISCS in the *terra incognita* (Section 4.1;
Juliano et al., 2022).


Much research remains to be done to continue to enhance our understanding of the scales of atmospheric motion
most relevant for harvesting wind energy. This team and the community have more work to do on the plethora of
complex cases. More research is needed to further improve coupling technologies. For instance, more research is
needed to understand why direct/indirect profile assimilation are successful in some cases and unsuccessful in

others. We should also continue to explore topics of complexity, both on shore and off shore. Much remains to be

learned through judiciously applying uncertainty quantification methods.

Although the current A2e MMC project has formally completed, we expect that its impact will live on, both in terms

of providing code and methodologies that can be used by a wide range of wind farm modelers and in terms of being

integrated into subsequent DOE wind energy projects. Specifically, DOE is initiating projects in offshore wind

energy, complex terrain modeling for wind energy, and the impact of extreme events on modeling for wind energy.

In deploying renewable energy, we have become more cognizant of issues of fairness and justice to the people being

impacted. In the United States, the Biden Administration's Justice40 Initiative (White House, 2022) seeks to deliver

40% of the overall benefits of climate investments to disadvantaged communities and inform equitable research,

development, and deployment within the DOE, has recently highlighted the importance for energy justice

considerations within the development of new energy systems. One of the major challenges of working in this space

is finding actionable, effective paths forward while acknowledging and respecting the existing legacy of

noninclusivity. Organizations such as the Initiative for Energy Justice and the Energy Equity Project (Initiative for

Energy Justice, 2022) have established guidelines for working in the space of energy justice. Specifically these

include: addressing the current perceptions that have been built on past practices; identifying uniquely

disadvantaged people; procedural fairness; making sure that access is equally tenable; making sure the quality of

service is equal across groups; and ensuring the desired impacts. Defined metrics can be used to determine whether

or not a project is successful in working toward energy justice. While fairly centered on policymaking, these

assessment points can help guide the focus of renewable energy development, and act as a compass for what

research objectives will have meaningful impact.

Finally, the MMC team wishes to thank colleagues and community members for input throughout the course of this

project. Our industry advisory panel and attendees to our various webinars and workshops have provided valuable

input as to the directions that we have chosen and solutions that may be most practical for application to real-world

needs. The biggest lesson learned is that it is through community cooperation that we are most likely to advance the

science and technology needed to deploy the amounts of wind energy that the world will need for a carbon-free

energy future.

**Author Contributions:** This paper results from a team effort to which all authors contributed. Project oversight and leadership plus management coordination was led by SEH with assistance from BK, LKB, CMK, JM, and MC. SD and MR provided oversight and resources from DOE, with MR providing overarching research goals and aims. EQ and PH led the software and data curation, with assistance from other authors. SEH led the preparation of the manuscript and all authors contributed.

**Competing Interests:** The authors declare that they have no conflict of interest.

**Acknowledgments:** This work was authored in part by the National Renewable Energy Laboratory, operated by Alliance for Sustainable Energy, LLC, for the U.S. Department of Energy (DOE) under Contract No. DE-AC36-08GO28308; Pacific Northwest National Laboratory (PNNL), operated by the Battelle Memorial Institute, for the

U.S. DOE under Contract No. DE-A06-76RLO 1830; and Lawrence Livermore National Laboratory, operated by
Lawrence Livermore National Security, for the U.S. DOE under Contract No. DE-AC52-07NA27344. Funding was
provided by the U.S. Department of Energy Office of Energy Efficiency and Renewable Energy Wind Energy
Technologies Office. The views expressed in the article do not necessarily represent the views of the DOE or the
U.S. Government. The U.S. Government retains and the publisher, by accepting the article for publication,
acknowledges that the U.S. Government retains a nonexclusive, paid-up, irrevocable, worldwide license to publish
or reproduce the published form of this work, or allow others to do so, for U.S. Government purposes. The National
Center for Atmospheric Research (NCAR) was a subcontractor to PNNL. NCAR is a major facility sponsored by the
National Science Foundation under Cooperative Agreement No. 1852977. The authors wish to thank the reviewers
whose comments and suggestions resulted in an improved manuscript.

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
