# Peer review of "Lessons learned in coupling atmospheric models across scales for onshore and offshore wind energy"

_Wind Energy Science, 2022_

## Author Comment (AC1)

**Response to Reviewers' Comments**

Reviewer 1: Javier Sanz Rodrigo

**General comments**
Excellent work summarizing a long research program on the complex topic of mesoscale-to-microscale coupling of atmospheric models for wind energy applications. The authors have covered a lot of ground exploring different techniques aiming at closing the gap between flow models that have been developed primarily by meteorologists using weather models on the mesoscale side and wind engineers on the microscale side using CFD codes. Hence, the challenge is not only scientific and technical but also sociological towards the development of a long-lasting interdisciplinary collaboration. Kudos to MMC for that!

The summary is quite comprehensive, given the complexity of the topic, with well-described challenges, core scientific topics and computational techniques established along a validation framework of case-studies of increasing complexity. I like the narrative of the research journey. If anything, the authors could try to mitigate some of the fluid dynamics jargon to improve accessibility to researchers from other neighboring disciplines.

Out of each chapter there are a number of lessons learnt that are collected in the conclusions with reference to all the papers published by the team. Since the topic is still far from being solved it is highly appreciated all the efforts being made by the MMC team to open-source much of the code and data through GitHub repositories and a documentation website. As the main deliverable of the MMC project I would try to publish these repositories properly by associating them with persistent DOIs to allow citation and to track usage and impact.

I have identified other topics in the text that I believe deserve being highlighted because of the practical benefits they bring. I've also raised a few questions to see if the MMC team has arrived to some consensus on the best method to produce high-quality input data for offline coupling to microscale. I believe this is a key objective. Since WRF is the de-facto standard mesoscale model for wind energy it would be worth having a reference setup that narrows down the configuration options and provide valid simulations in the terra incognita and/or consistent input data to drive microscale models. I believe it is the right time to make explicit all the guidelines that can help reducing the spread of model configurations so research efforts can be focused on the key drivers of uncertainty for wind energy. As it is discussed in section 2.5, much work remains on uncertainty characterization by blending high-fidelity modeling and data-driven ML approaches. Is coupling to data *the next terra incognita*? I believe so…

We thank the reviewer for the complementary comments in the General Comments section. In our final proofing, we did take into account the comment about avoiding jargon and worked toward that goal. We also took the reviewer's suggestion to obtain persistent DOIs. We have also archived case setups to demonstrate those that have been shown to work well. We have also tried to make guidelines explicit where we feel comfortable doing so, but wish to be careful to not be overly definitive where we believe further research is merited before making those

recommendations. There are also some excellent specific suggestions and responses to those are included in the Responses to Specific Comments in blue text below.

**Specific comments**

53: Please provide a bit more context about what the Boussinesq approximation stands for. The sentence is modified to include a parenthetical brief definition. "For instance, the mesoscale models are fully compressible while microscale models are typically incompressible or Boussinesq (density differences are ignored except due to buoyancy)."

54: Please provide more context on why the surface conditions are inherently different. The MOST surface boundary condition is derived on the basis of relating average surface fluxes to corresponding average values of relevant atmospheric parameters (winds, temperature, moisture), under steady (in time) forcing conditions, and homogeneous (in space) surface cover, temperature, moisture, etc. This relationship holds reasonably well in applications that prognose ensemble mean values of the atmospheric flow parameters, i.e. Reynolds Averaged Navier Stokes (RANS), such as mesoscale weather simulations, and moreover for applications over nearly homogeneous surfaces. However for LES, which uses sufficient resolution to capture small-scale features of both the environment and the flow, applicability of MOST is not on solid theoretical grounds. This is true even under steady, homogeneous conditions, due to the explicit resolution of the rapidly evolving, small-scale features within the flow variables that violate the ensemble mean basis of MOST. Its applicability is degraded further when the high resolution used in LES is also able to capture small-scale surface heterogeneities. For these reasons, fundamentally different approaches are required for LES applications of the surface boundary condition (and even for high-resolution unsteady RANS applications). For our purposes in this bullet, we added the phrase, "due to surface inhomogeneities that become important at the microscale." Note that we now also point the reader to section 2.2 as suggested by Reviewer 2.

90: Although this is defined in detail later, can you define what online vs offline mean since these terms are used in many other domains? A brief description of each has been added directly to the sentence, which now states, "Several microscale models have been tested, including the large-eddy simulation (LES) version of WRF (WRF-LES) that can be run online where the inner nest derives the conditions directly from the outer nest during the simulation, and several offline models, which are run after the mesoscale model with inputs derived from those previous runs."

93: dealing with (flow?) complexity. This sentence has been modified to state, "…and dealing with multiple sources of flow complexity, including complex terrain, coastal flows, and offshore flows."

111: I would add a reference to the public repository and associated documentation in the References section and then use citations in the text. In fact, you may use Zenodo's GitHub integration to archive the repo and automatically issue a new DOI each time you create a new GitHub release. https://docs.github.com/en/repositories/archiving-a-github-repository/referencing-and-citing-content

Thank you for this suggestion, which we have taken. The text now refers to permanent github directories created via Zenodo.

121:  presnts > presents   Fixed

148: "In some situations": can you be more specific? The wording has been modified to, "In situations when the GRR (between the mesoscale and microscale domains) becomes large, it can be beneficial to use the LES three-dimensional (3D) turbulence model (e.g., Smagorinsky, 1963) in the *terra incognita* region, …"

135: The MMC team has correctly focused on a single mesoscale code (WRF) to explore its use in the "terra-incognita". This has resulted in some guidelines and case-study setups that constitute valid benchmarks to improve upon in the future. Would it be worth highlighting this deliverable of the project? Is there a reference WRF-LES setup that can be used by the wind industry to run meso-micro simulations at ~100 m resolution? This version-controlled benchmark model will be used to quantify improvements going forward as the model is tested in other sites subject to different wind climates and terrain complexity than those used so far by the MMC project. Such reference (namelist) configuration files are made available in the code repository (section 2.7) but it could be provided as an Annex of this paper to highlight this release in a similar way as it was done in the New European Wind Atlas WRF model (https://doi.org/10.5194/gmd-13-5053-2020).
We have taken this suggestion and created an Annex to the paper that is consistent with the permanent doi's.

226: This categorization of coupling strategies is very comprehensive. It would be worth highlighting this with a chart or a table so this can stand out from the rest of the text of this section.
Figure 2 has been added to satisfy the request - we agree this helps to make the classification clearer.

250: What about using heterogeneous mesoscale tendencies (3D instead of 1D)? Has this option been discarded by the group for any reason or simply not tested? This approach was followed in the Alaiz benchmark showing added value of 3D vs 1D tendencies in the prediction of mean wind speed profiles in complex terrain especially under stable conditions (https://iopscience.iop.org/article/10.1088/1742-6596/1934/1/012002).
Text has been added to the manuscript referring to this paper, "Although it is theoretically possible to apply an internal source term that varies three-dimensionally in space to represent horizontally heterogeneous situations, we have not explored that approach; however, others (Sanz-Rodrigo et al., 2021) have demonstrated the validity of that approach.  Instead, for horizontally heterogeneous domains, or simulations that resolve turbines, we have focused our attention on boundary-coupled simulations, which provide the highest degree of generality."

262: "mesoscale meteorology models are usually not developed with LES applications in mind". Would the authors suggest putting more efforts into improving computational efficiencies in the meso-micro (WRF-LES) range or, as the text suggests, would the MMC team rather focus those

efforts on improving the offline coupling approach to benefit from the greater flexibility and throughput of microscale models?

This is a great question that may occur to other readers as well. We added a couple of sentences to address this: "Note that a current DOE initiative focuses on development of mesoscale (ERF) and microscale (AMR-Wind) models that are aimed at exascale HPC platforms. However, also note that online coupling of mesoscale and microscale models that are based on the same formulation, i.e., equations, and use the same numerical discretization simplifies coupling and results in more consistent simulations across scales."

264: Offline boundary coupling: Is there consensus on what are the best practices for the mesoscale model to provide consistent inputs to microscale models (e.g. resolution, domain size, etc)? Would the same guidelines for online coupling (nesting) hold for offline coupling? It would be worth making these guidelines explicit.

Although it would be nice to give explicit guidelines, we are not ready to do that. More research is needed. Although some general principles may be consistent between online and offline methods, the best practices may not match.

282: To provide some perspective on the magnitude of the computational burden, can the authors provide an estimate of the computational overhead due to the additional fetch required to spin up turbulence? (e.g. with respect to a simulation with periodic boundary conditions)

Quantitative general statements about reduction of fetch are unfortunately quite difficult, due to multiple contributing factors, including surface roughness and terrain, wind speed, and atmospheric stability. The references cited within the manuscript do provide quantitative assessments of the reduction of computational overhead for particular forcing conditions, while also showing the general result that the reduction of fetch required to spin up turbulence to within some percentage of its equilibrium value increases with both static stability and with mean wind speed. The former is due to turbulence generation having to compete with buoyancy suppression, while the latter owes to the rapid advection of the flow through the computational domain during the turbulence development process. Very broadly speaking, for a computation using specified inflow conditions during unstable conditions, the reduction of fetch due to perturbations can be small, perhaps only around 100 grid cells in the direction of the mean flow. However, during neutral or stable conditions, perturbation can foreshorten the fetch by several hundred grid points, which can constitute a computational savings of 50% or more. A shortened version of this explanation has been added to the text of the manuscript.

To answer the second part of the inquiry, the turbulence spinup process in a domain using periodic lateral boundary conditions is different on a practical level, from one using specified inflow. In the case of periodic conditions, the simulation time required to approach equilibration of turbulence is the relevant factor, as the perturbations are applied at the beginning of the simulation only. The size of the computational domain is largely irrelevant, as long as it is large enough to contain the largest turbulence structures, since the flow can exit and re-enter the domain as many times as is required. For domains using specified inflow conditions, however (e.g., from a mesoscale simulation), the flow only remains within the computational domain for the duration of the advective timescale of the flow across its extent (length of domain/wind speed). Therefore, turbulence has to be generated entirely within the computational domain, and quickly enough to be adequately developed before advecting to the region of interest.

341: Can you provide references about the parameter ensemble approach and meta-modeling techniques?
Several references have been added.

400: This paragraph is not specifically addressing complex terrain. It rather discusses the advantages of using a microscale model. Maybe it could be moved to section 2.3.
This paragraph has been moved to section 2.3 as suggested.

411: "or sometimes it may be of secondary importance" I would remove this statement since it is already implicit in the previous one.
This phrase has been deleted as suggested.

420: While the Rayleigh damping method is relatively simple, based on two parameters, section 2.6.2 suggest that the selection of these parameters is difficult to judge a priori involving fine tuning to the specific site. Are there some guidelines that could be provided? Again, making these explicit would be great even if some fine tuning would still be necessary.
We have added text, " We suggest a damping layer thickness of 3-5 km with a damping time constant of 0.005 1/s, but additional tuning likely will be required."

435: SST gradients: I guess this argumentation is also valid for onshore conditions when we may need to account for surface temperature heterogeneity in complex terrain or across different land-use patches. Is this a particular challenge for offshore because its relative importance in the flow field is greater than in onshore conditions?
Yes, this would be valid for onshore surface temperature heterogeneity as well. We believe (and may need to look into this further) that the big issue is that, unlike on the land surface, SST gradients are constantly changing. We rely heavily on remote sensing to capture these gradients but those are often available on only a  daily frequency and they are frequently obscured by clouds or simply not captured well due to the granularity of the remote sensing equipment. Thus, accurately capturing the surface forcing is very difficult offshore.

452: "very complex modeling framework requiring the coupling of ocean and atmospheric models". Not sure if atmospheric here would only relate to mesoscale models (probably not). Would the authors suggest that the main complexity resides in the mesoscale range such that we could essentially use the same microscale model setups that we used in flat-terrain onshore conditions? Is there any particularity on surface boundary conditions for microscale models to mimic different ocean conditions? Can we still use MOST? I guess these questions will be addressed in future MMC offshore-focus research projects but it would be nice to elaborate a bit on perspectives for microscale models.
This is somewhat tough to answer... in the end, we did use essentially the same microscale setup as we would for onshore, flat terrain. However, to say that the complexity lies in just the mesoscale for the setup isn't really true for onshore, flat terrain (perturbation methods, grid refinement for stable conditions, dealing with terra incognita, etc.). Then, because of ocean state variables, the air-sea interface is a very complex environment with a lot of study surrounding it (waves, SST gradients, etc.). For this setup, we also ended up using MOST, but at the same time were working on a machine learning surface layer for offshore to improve upon MOST, which

will be described in a forthcoming paper. The reviewer is correct that many of these issues are currently being studies in a follow-on offshore modeling project where we hope to define some of these issues better. We don't feel ready to discuss these in the current paper, however. Thanks for the perceptive comment.

455: It is great to make the MMC project open source. As mentioned earlier, the only missing piece is to have the repositories associated to DOI so they can be cited appropriately. Otherwise, it's all very comprehensive and well-structured for anyone to follow and contribute.
Thank you. As mentioned previously, we have now obtained persistent DOIs and changed the citations in the text accordingly. Thank you for that suggestion.

505 (and elsewhere): I would move the link to the reference list as a website-type publication.
This referencing style for web addresses has been modified as suggested here and throughout the manuscript.

520: The GABLS3 case can be mentioned as an alternative to study a flat terrain diurnal cycle with high-altitude measurements (1000m+).
  * https://doi.org/10.5194/wes-2-35-2017 (meso-micro paper)
  * https://doi.org/10.5281/zenodo.834355 (repository)
  * http://iopscience.iop.org/article/10.1088/1742-6596/854/1/012037 (benchmark results)
Thank you for pointing out this omission of a reference to an important case study in the literature that our team members participated in. Reference to this set of resulting publications has been added at the end of this section. Note that when we began the SWiFT case early in our project, DOE headquarters was insistent that we use data from their SWiFT site for our case study, despite the fact that it is a scaled facility.

565 (Figure 2): For completeness, can you provide a definition of lamda? A definition of lamda has been added in the paragraph before the figure.

635 Figure 4: Are these integral lengths scales all calculated at the same height (same as in Figure 3)? Please specify in the caption for completeness.
The text and figure caption have been modified to clarify that we used multiple coupling methods, but used a single method to compute the integral length scale at 80 m.

703: I assume that all domains are running on the same vertical levels. Please confirm and provide some indication about horizontal and vertical resolutions to get some perspective about the transition from mesoscale to microscale across the different domains.
Yes, all results shown here used 88 vertical levels with 20 m spacing below 1 km. This information has been added to the manuscript.

710 Figure 8: what is "error"? The text says "ensemble mean error" but it is not clear which simulations form the ensemble and which metric is it (MAE, RMSE, etc). Please clarify and add units to the variables.
The equation for ensemble mean error is now included in the text. Units for Fig. 9 (previously Fig. 8) are included in the figure caption.

743: Simulations with 250 m resolution using the 3D PBL Mellor Yamada are still called mesoscale. Is this because the MMC team is trying to bring mesoscale inputs closer in resolution to the LES domain as introduced in section 2.1? What does this mean in practice? Is this 3D PBL scheme enabled in all mesoscale domains? What is the computational overhead vs traditional 1D PBL? Can this allow (much) smaller LES domains to improve computational efficiency or do we still need large LES domains to allow meso-micro turbulence to develop? I'm just trying to understand if 2D PBL is part of the WRF-LES strategy or simply meant to be used as a mesoscale-only approach to increase resolution and improve accuracy in complex terrain sites.

We chose to test the 3D PBL in the terra incognita specifically to determine to what extent it alleviates the problem of spurious waves. We have modified the text in Section 4.1 to include some of these points: "This grid spacing is considered to be mesoscale resolution because it is not fine enough to fully resolve the most energetic eddies (i.e., the LES limit) due to the model's effective resolution. The three single domain, doubly periodic configurations are: homogeneous surface forcing (rolls and cells), sea breeze front initiation, and mountain–valley circulation. Results clearly depict the suppression of M-CISCs by the 3D PBL scheme compared to a traditional 1D PBL scheme (Juliano et al., 2022). The impact of the turbulent length scale/closure constant's formulation is found to be very important, such that M-CISCs may be present in the 3D PBL solution when the length scale is insufficiently large and thus vertical mixing is not strong enough. In general, we believe that the 3D PBL parameterization has potential to be useful both as a mesoscale-only approach and as part of a mesoscale-microscale coupling strategy."

785: Please define the variables in the Figure. What does "Ug and Vg only" means?
The variables are now defined in the caption.

**Reviewer 2:**

**General comments**

The authors present the major findings that have been made by the Mesoscale to Microscale Coupling (MMC) team of the A2e initiative of the U.S. Department of Energy in the last couple of years.

Different approaches of coupling mesoscale and microscale models have been investigated by the team and applied to cases with step and step increasing complexity. One major achievement of the efforts made by the MMC team is the development of methods that help to overcome the long-standing problem of the terra incognita in atmospheric modelling. Moreover, the capabilities of several methods to ensure the development of turbulence in the microscale simulations coupled to the mesoscale ones are assessed.

A fast transfer of the knowledge gained by the MMC team to all modellers in the wind energy community as well as follow-up research by an even broader community is supported by the fact that the MMC team has made the simulation codes as well as pre-processing and postprocessing tools for several case studies publicly available on GitHub. Online documentation is also provided.

Summarizing, the paper presented by the members of MMC team is a great presentation of great work that will help to improve the modelling of wind resources and therefore support the further deployment of wind energy. I have only a couple of minor comments and therefore, I'm supporting the acceptance of the manuscript for publication in Wind Energy after minor revisions.

We thank the reviewer for the complimentary comments in the General Comments section.

**Specific comments**

Line 49/50: „Thus, the solution to obtaining accurate flow prediction representing all relevant scales is to couple the models at these scales." What is meant by „these scales" here?
This sentence has been reworded to be more explicit, "Thus, the solution to obtaining accurate flow prediction representing all relevant scales is to couple the mesoscale model to the microscale model."

Paragraph starting in line 52: My suggestion is to present the issues with coupling mesoscale and microscale models that are mentioned here in the form of a list with bullet points (similar to the list of objectives that is presented starting in line 69). Moreover, there could already be links made to those subsequent chapters where these issues are further addressed.
Excellent suggestion. This has now been done.

Line 51/52: Here, the problem of compressible mesoscale but incompressible microscale models is mentioned. Where is this problem later be addressed in the text?

This issue was not explicitly addressed in the work described here, but rather merely noted as a difference that must be considered. We suspect that it may be part of the issue with spurious gravity waves triggered in some instances of forced flow, such as described in section 2.6.2. We have not studied this sufficiently, however, to make definitive statements.

Line 90: To make the contents of the paper accessible to a broad community the meaning of the terms offline and online in this context should be explained.
This is now done and the manuscript now reads, "Several microscale models have been tested, including the large-eddy simulation (LES) version of WRF (WRF-LES) that can be run online where the inner nest derives the conditions directly from the outer nest during the simulation, and several offline models, which are run after the mesoscale model with inputs derived from those previous runs. "

Line 101/102: To increase the clarity of subsequent descriptions I suggest to add „internal coupling" in brackets at the end of the sentence starting in line 101. Done

Line 102/103: For the same purpose my suggestion is to add "external coupling" in brackets at the end of the sentence starting in line 102. Done

Line 122: presnts --> presents Fixed

Line 275: generate --> generated Fixed

Line 282/283: "the flow field within the fetch will not represent either the mean and turbulence fields during the process of turbulence spin-up and equilibration" Is this a specific problem of the spin-up time only? Close to the inflow boundary of the microscale domain (a region that I would understand as being in the fetch) even after the spin-up of turbulence these issues remain, don't they?
In the region near the lateral inflow plane(s) of an LES domain, where mesoscale inflow that does not contain resolved-scale turbulence is provided to the LES domain, there is indeed a fetch region within which neither the mean flow nor the turbulence field is accurately represented. Within the fetch region, both the turbulence and mean flow statistics change rapidly, with turbulence developing, and the mean flow responding to those changes. Random perturbations applied just inside the inflow plane(s) produce uncorrelated gradients that, through the action of the governing equations, develop into robust turbulence features with expected correlations and energetics. During this process, there is often an associated reduction in mean wind speeds and a small change in wind direction near the surface, due to a temporary reduction in downward momentum transport -since the mesoscale closure is no longer providing that within the LES domain, and the turbulence within the LES domain has not yet developed the correlated structures responsible for downward momentum transport.

The studies that have examined this process in detail have shown that the length of this region varies with stability and mean wind speed, with more stable and higher wind speeds generating longer transitional fetches. However, the mean and turbulence statistics of the flow do asymptotically approach their equilibrium values, after which no significant changes are observed with increasing distance from the inflow. Moreover, these flow statistics also approach

those of separate LES conducted using the same forcing conditions, but with periodic lateral boundary conditions, which provide reliable quasi-equilibrium statistics, and are, as such, often taken to be the "truth", from the perspective of what the nested domain should converge to given an infinite fetch.
This explanation seemed too long to add to the text itself, but we have now added a footnote to convey this information.

Line 352-354: Please add an information which of the two cases that are presented showed a larger sensitivity of the modelled flow on the eddy viscosity coefficient.
This section has been augmented and now reads, " However, the sensitivity of the modeled flow to variations in this parameter was noted to vary significantly between two case studies with nominally similar large-scale flow conditions but different smaller-scale flow structures (convective cells versus rolls), and to show nonlinearity of response. For example, the hub-height wind speed showed much greater sensitivity to the eddy viscosity coefficient, across the full range of  eddy viscosity coefficient values that were tested, in the case with roll-type structures. TKE was also more sensitive in the case with rolls to changes in the coefficient value through the lower half of the range of values tested. At higher values of the coefficient, turbulence was effectively damped, so that the sensitivity of TKE to further increases in the coefficient became slight. In contrast, the case with a cellular flow structure was better able to sustain turbulence, so sensitivity of TKE to the eddy viscosity coefficient persisted across the full range of tested values, and sensitivities were greater at higher values of the coefficient."

Line 369: atmophseric --> atmospheric  Fixed

Line 391: How large are the fetch requirements? How strong is the fetch length requirement reduced in the case of applying a perturbation technique? What does the required fetch length depend on? Some quantitative statements would be valuable. Moreover, it would be interesting to suggest methods that help to assess whether the fetch length is sufficient or not.
In flat terrain under neutral stratification fetch is virtually infinite as we have shown in several papers, starting with Mirocha et al. 2014 (MWR, https://doi.org/10.1175/MWR-D-13-00064.1). In that paper a weakly convective case was studied too, and without perturbations turbulence does not develop when dissipative subgrid models are used. Turbulence develops after a 10 km fetch when the nonlinear anisotropy backscatter model is used in WRF. Reference has been added to that paper.

General comment concerning chapters 3 and 4: My suggestion is to add also references to the papers from which results are presented in the titles of the subsections. To have that direct link would prevent that the reader needs to go back to chapter 2 to find that information.
Thank you for this comment. Adding references to subtitles doesn't fit the referencing requirements of this journal.  We have been careful in this section, however, to reference the papers that have more details of the results. Note that for some of these sections, however, the more detailed papers are still in preparation.

Line 486: feature --> features  Fixed

Line 614/615: „The larger extent allowed a fetch for turbulence development." What size of the fetch is required in order to get fully developed atmospheric turbulence?
The following text has been added to the manuscript, "Convergence of vertical profiles of turbulent metrics was observed within a 3 km fetch distance. Thus, all the boundary-coupled scenarios considered were set up with a large 3 km extent fetch region to allow turbulence development."

Line 614: It would have been interesting to use the same horizontal extension of the model domain also for the SOWFA-IPA case in order to be able to exclude that the size of the model domain impacts the results obtained.
The first 3 km of the SOWFA CPM and all the WRF simulations were not considered in the analysis because the turbulence was still developing in that region. For the SOWFA IPA method, due to its periodic boundary conditions, we have developed turbulence everywhere. The goal in setting these domains of different size was to explicitly account for only 3 km of developed-turbulence extent. If much larger structures were allowed to be developed in the SOWFA IPA case (and that case only), that could skew the results of the integral scales. The manuscript was modified to clarify this.

Figure 3: In addition to this figure, it would be interesting to show figures of the turbulence spectrum. How differently distributed is the turbulence energy in these different cases? How do the spectra change with position inside the domain?
A figure has been added to demonstrate the turbulence spectrum of the different models as suggested. The referenced paper includes more complete information with spectra plotted at different times within the 4-hr interval. The text has been augmented with, "Figure 5 shows the energy spectrum during one hour of the 4-hour period of interest. The spectrum was obtained using 10-min Welch windows with a 50% overlap, considering an ensemble average of several locations within the 3 x 3 subdomain shown in Figure 4 (previously Figure 3), leveraging horizontal homogeneity."

Line 659: In order to be able to assess whether a domain with an extension of 6 km x 6 km provides a large fetch it would be helpful to provide information on the fetch length that is actually required to get fully developed atmospheric turbulence.
As the cited papers show, the turbulence asymptotically approaches equilibration over a computational distance of a few hundred grid cells in similar setups to the one used here. Moreover, the location and time period of the simulations shown in what is now Fig. 7 (previously Figures 5 and 6) are similar to the results shown in Fig. 4 (previously Figure 3), which use the same LES domains (6 km with 600 grid cells in each horizontal direction for a horizontal resolution of 10 m, with the turbine in Figures 7 located approximately in the middle). Figure 4 shows the turbulence appearing to have nearly equilibrated within the subset of the LES domain in question. The manuscript now clarifies that the domain setup matches that of the previous section, which discussed these issues.

line 665: trubulence --> turbulence  Fixed

Figure 5: The font (size and type) used in subparts a) and b) of this figure should be identical.
Jeff

Figure 6: Please apply axis labels.
We have chosen to delete this figure and refer to a forthcoming paper that will compare the WRF-SOWFA-GAD approach to the one used in WRF-LES-GAD portrayed in what is now Fig. 7 (previously Fig. 5). We decided that the two figures should be comparable, but were not able to rerun the simulation in time to include in this paper.

Line 683: My suggestion is to present the details on the different SST data sets used rather in a table instead of in the text.
We have taken this suggestion and added Table 1 in place of the verbiage in the original manuscript.

Line 688: I think this is actually the first time that an information on the grid spacings applied is given. This would also be an interesting information for the other cases presented. This would mean a homogenization of the presentation of the different studies presented.
We have worked to do that where that information is readily recoverable, but due to the project completing, unfortunately some of the information was difficult to recover.

Line 731: Smagorsinky --> Smagorinsky      Fixed

line 750: atmospheric boundary --> atmospheric boundary layer  Fixed

Figure 10: What is the meaning of the shaded areas in this figure? Do all subfigures apply the same legend? In my opinion the figure showing the skin temperature can be deleted.
The figure caption now describes the meaning of the shaded areas, which are in terms of a standard deviation. We retained the skin temperature plot because it is the boundary condition driving the sensible heat flux.

Figure 11: What is actually shown in the upper part of this figure and where is the reference in the text to this? What is the meaning of the third line presented in the subfigures in the bottom half of this figure?
The caption for Figure 11 (now Figure 12) has been greatly enhanced.

Section 4.3: Please clarify: what is the temporal resolution of the 960 m resolution input field and the resulting GAN generated 30 m resolution field? Is the refinement of the data done in space and time? Or only in space? If the refinement is also done in time how well do the turbulence spectra agree with those of a full LES run?
The 960 m input WRF files were generated every 3 min. The resulting GAN 30 m is the same temporal resolution as the input data. The refinement of data is only accomplished in space and there is no time component in this application. It merely ingests a tile for a given time and downscales it for the same time. We have worked to clarify this in the opening paragraph.